# Highly parallel optimisation of chemical reactions through automation and machine intelligence

Joshua W. Sin[1,2] ✉, Siu Lun Chau[3], Ryan P. Burwood[4], Kurt Püntener[1], Raphael Bigler[1] & Philippe Schwaller [2,5] ✉

We report the development and application of a scalable machine learning (ML) framework (Minerva) for highly parallel multi-objective reaction optimisation with automated high-throughput experimentation (HTE). Minerva demonstrates robust performance with experimental data-derived benchmarks, efficiently handling large parallel batches, high-dimensional search spaces, reaction noise, and batch constraints present in real-world laboratories. Validating our approach experimentally, we apply Minerva in a 96-well HTE reaction optimisation campaign for a nickel-catalysed Suzuki reaction, tackling challenges in non-precious metal catalysis. Our approach effectively navigates the complex reaction landscape with unexpected chemical reactivity, outperforming traditional experimentalist-driven methods. Extending to industrial applications, we deploy Minerva in pharmaceutical process development, successfully optimising two active pharmaceutical ingredient (API) syntheses. For both a Ni-catalysed Suzuki coupling and a Pd-catalysed Buchwald-Hartwig reaction, our approach identifies multiple conditions achieving >95 area percent (AP) yield and selectivity, directly translating to improved process conditions at scale.

Chemical reaction optimisation is a challenging and resource-intensive yet essential process in chemistry. Chemists explore combinations of various reaction parameters (e.g., reagents, solvents, catalysts) to simultaneously optimise multiple objectives such as yield and selectivity. In process chemistry, reaction optimisation faces more rigorous demands on reaction objectives than in academic settings, encompassing additional economic, environmental, health, and safety considerations[1,2]. These factors often necessitate the use of lower-cost, earth-abundant, and greener alternatives, such as replacing traditional palladium catalysts with nickel[3,4], and selecting solvents that adhere to pharmaceutical guidelines[5]. Optimal reaction conditions satisfying these stringent criteria are often substrate-specific and challenging to identify generally for a given set of reactants.

Advancements in automation[6–11] have catalysed a shift in chemical reaction optimisation[12], particularly through the emergence of chemical high-throughput experimentation (HTE) adapted from techniques in biology[13]. HTE platforms, utilising miniaturised reaction scales and automated robotic tools, enable highly parallel execution of numerous reactions. This allows for the exploration of many combinations of reaction conditions, making HTE more cost- and time-efficient than traditional techniques relying solely on chemical intuition and one-factor-at-a-time (OFAT) approaches[12,14]. However, as additional reaction parameters multiplicatively expand the space of possible experimental configurations, exhaustive screening approaches remain intractable for larger design spaces, even with HTE[14]. Consequently, HTE chemists rely on chemical intuition to navigate vast

[1]Process Chemistry & Catalysis, Synthetic Molecules Technical Development, F. Hoffmann-La Roche AG, Basel, Switzerland. [2]Laboratory of Artificial Chemical Intelligence (LIAC), EPFL, Lausanne, Switzerland. [3]Rational Intelligence Lab, CISPA Helmholtz Center for Information Security, Saarbrücken, Germany. [4]Solid State Sciences, Synthetic Molecules Technical Development, F. Hoffmann-La Roche AG, Basel, Switzerland. [5]National Centre of Competence in Research (NCCR) Catalysis, EPFL, Lausanne, Switzerland. ✉e-mail: wing_pong.sin@roche.com; philippe.schwaller@epfl.ch

reaction spaces effectively. A common approach is designing fractional factorial screening plates with grid-like structures[8,15] (Fig. 1a), as used in our process chemistry HTE lab[16]. While these structures effectively distill chemical intuition into plate design, they explore only a limited subset of fixed combinations. This limitation, especially in broad reaction condition spaces, may lead traditional approaches to overlook important regions of the chemical landscape.

From advances in computer science and statistics[17,18], machine learning (ML) techniques, particularly Bayesian optimisation, have gained popularity in chemistry for their ability to successfully guide experimental design[19–21]. Bayesian optimisation uses uncertainty-guided ML to balance exploration and exploitation of reaction spaces, identifying optimal reaction conditions in only a small subset of experiments. Bayesian optimisation has shown promising results in reaction optimisation, validated experimentally by multiple case studies[14,22–24] and outperforming human experts in simulations[20]. However, existing applications in reaction optimisation have been largely limited to small numbers of experiments in parallel batches of up to sixteen[14,20,22–24]. Moreover, these approaches are often non-automated or are restricted to single reaction objectives. The constraints of small batch sizes necessitate more optimisation iterations to identify high-performing reaction configurations, particularly when exploring large reaction spaces. These limitations also impede integration with large-scale automation. Consequently, the full potential of highly parallel, automated reaction optimisation remains under-explored. In the pharmaceutical industry, where rapid development is crucial, many reactions prove unsuccessful[25,26]. There is a pressing need to expedite optimisation strategies in chemical synthesis beyond traditional approaches to meet increasingly demanding timelines in drug discovery and development. The natural synergy between ML optimisation and HTE platforms, leveraging efficient data-driven search strategies with highly parallel screening of numerous reactions, offers promising prospects for automated and accelerated chemical process optimisation in minimal experimental cycles.

In this work, we report the development and application of an ML-driven workflow for scalable batch reaction optimisation of multiple reaction objectives, applicable to any reaction of interest (Fig. 1b). Through in silico studies, we showed the scalability of our approach in handling large batch sizes of 96 and high-dimensional reaction search spaces of 530 dimensions. We also assessed the robustness of our workflow to chemical noise and accommodated for batch constraints typical in chemical laboratories, demonstrating its suitability for real-world applications. We applied our approach experimentally through an automated 96-well HTE optimisation campaign for a nickel-catalysed Suzuki reaction, exploring a search space of 88, 000 possible reaction conditions. Our optimisation workflow identified reactions with an area percent (AP) yield of 76% and selectivity of 92% for this challenging transformation, whereas two chemist-designed HTE plates failed to find successful reaction conditions. This study demonstrated advantages of our approach over traditional, purely experimentalist-driven methods, highlighting its potential to accelerate automated reaction optimisation. We further validated our approach through two pharmaceutical process development case studies. Our ML workflow rapidly identified multiple reaction

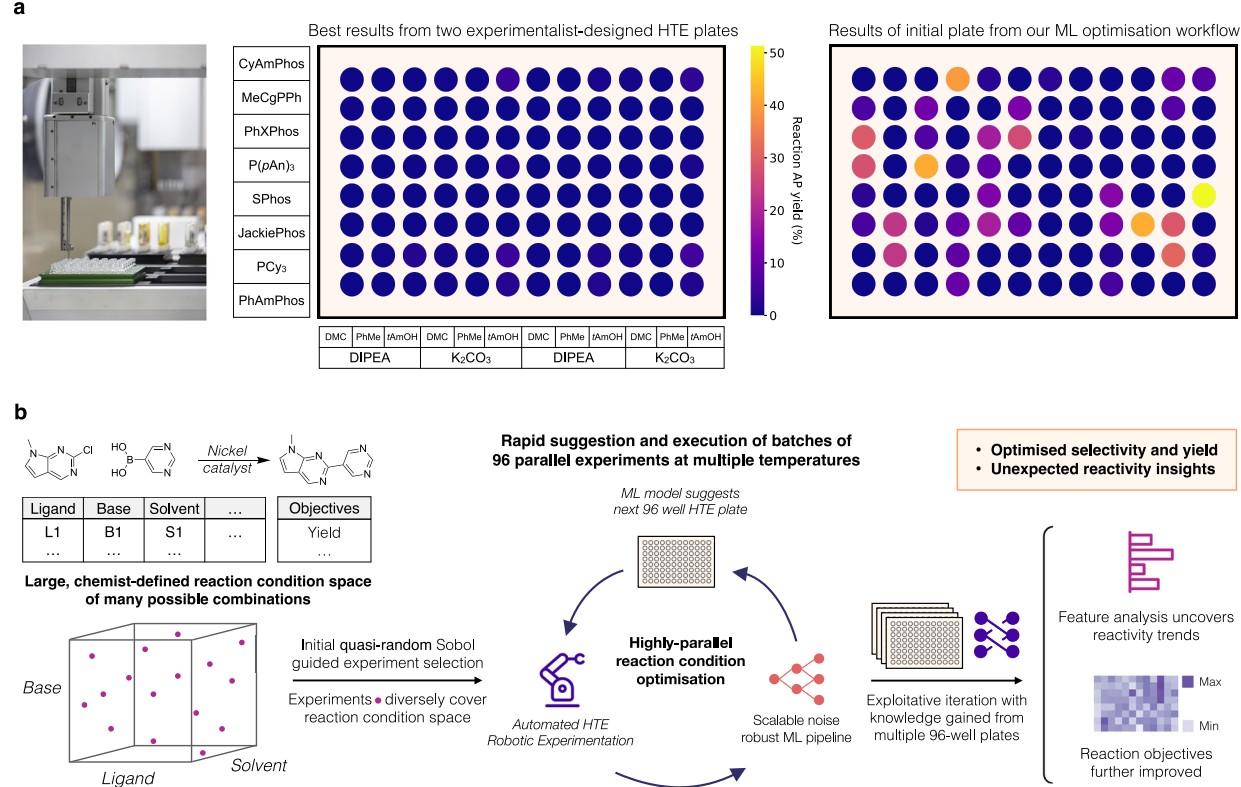

**Fig. 1 | Strategies for high-throughput experimentation (HTE) reaction optimisation and overview of this study. a** Comparing methods for HTE plate design. An example of a HTE plate designed with traditional fractional factorial grid-like structures and an initial HTE plate for our machine learning (ML) optimisation workflow. **b** Experimental application of our ML optimisation workflow to a Ni-catalysed Suzuki reaction. The experimentalist first defines promising reaction parameters (e.g., ligands, bases, solvents) comprising the reaction condition search space. The initial experiments are selected with quasi-random Sobol sampling[28], diversely sampling from the reaction condition space. Then, iterative Bayesian optimisation suggests subsequent HTE screening plates, optimising towards experimentally defined objectives. This process is typically repeated until convergence, stagnation in improvement, or exhaustion of experimental budget. Then, a fully exploitative ML approach leverages accumulated data from all HTE plates in the campaign to maximise final reaction objectives. Finally, feature analysis elucidates reactivity trends in the optimisation campaign.

conditions achieving >95 AP yield and selectivity for both a Ni-catalysed Suzuki coupling and a Pd-catalysed Buchwald–Hartwig reaction, significantly accelerating process development timelines. In one case, our ML framework led to the identification of improved process conditions at scale in 4 weeks compared to a previous 6-month development campaign. The 1632 HTE reactions conducted in this study are made available in the Simple User-Friendly Reaction Format (SURF)[27] with the custom code used in an open-source code repository, Minerva.

## Results

### Overview of optimisation pipeline

In our optimisation process, often involving reactions with sparse historical data, we prioritised thoroughly exploring a large set of categorical variables. From chemical experience, categorical variables such as ligands, solvents, and additives can substantially influence reaction outcomes, potentially creating distinct and isolated optima in the reaction yield landscape. Algorithmic exploration of these categorical variables enables the identification of promising reaction conditions, which leads to further refinement of parameters such as catalyst loading and activation in later stages of the optimisation process. However, incorporating numerous categorical parameters increases the dimensionality and complexity of the search space, as molecular entities must be converted into numerical descriptors, unlike directly representable continuous variables. We represented the reaction condition space as a discrete combinatorial set of potential conditions. These conditions comprised reaction parameters such as reagents, solvents, and temperatures deemed plausible by a chemist for a given chemical transformation, guided by practical process requirements and domain knowledge (Fig. 1b). This allowed for automatic filtering of impractical conditions such as those with reaction temperatures exceeding solvent boiling points or unsafe combinations like NaH and DMSO.

We then initiate our ML-driven Bayesian optimisation workflow with algorithmic quasi-random Sobol sampling to select initial experiments, aiming to sample experimental configurations diversely spread across the reaction condition space[28] (Fig. 1b) (see Methods). By maximising reaction space coverage of the initial experiments, Sobol sampling increases the likelihood of discovering informative regions containing optima. Using this initial experimental data, we train a Gaussian Process (GP) regressor[29] to predict reaction outcomes (e.g., yield, selectivity) and their uncertainties for all reaction conditions. An acquisition function, balancing between exploration of unknown (uncertain) regions of the search space and exploitation of previous experiments, then evaluates all reaction conditions and selects the most promising next batch of experiments (see Methods for more details). After obtaining new experimental data, the chemist can choose to repeat this process for as many iterations as desired (Fig. 1b), usually terminating upon convergence, stagnation in improvement, or exhaustion of experimental budget. Throughout the campaign, the chemist can integrate evolving insights from each iteration with domain expertise and fine-tune the exploration-exploitation balance, or align the strategy to meet process specific development timelines.

### Scalable multi-objective acquisition functions

In real-world scenarios, chemists often face the challenge of optimising multiple reaction objectives simultaneously, such as maximising yield while minimising cost. HTE campaigns, characterised by larger batch sizes and range of reaction parameters, further amplify optimisation complexity. Computationally, scaling the parallel optimisation of multiple competing objectives towards high batch sizes is challenging and incurs considerable computational load[30]. For example, the q-Expected Hypervolume Improvement (q-EHVI) acquisition function[31], a popular multi-objective acquisition function applied previously in reaction optimisation[14], has time and memory space complexity scaling exponentially with batch size[32]. Given the limited scalability of such approaches (see Supplementary Information Section 2), we sought to develop a more scalable optimisation framework for highly parallel HTE applications, incorporating several scalable multi-objective acquisition functions in our work: q-NParEgo[32], Thompson sampling with hypervolume improvement (TS-HVI)[30], and q-Noisy Expected Hypervolume Improvement (q-NEHVI)[32] (see Methods for further technical details).

### Benchmarking and evaluating

To assess optimisation algorithm performance, practitioners often conduct retrospective in silico optimisation campaigns over existing experimental datasets[14,20,33,34]. This enables comparison of optimisation performance against previously established experimental optima within a set evaluation budget. However, publicly available experimental datasets usually contain only ~1000 reaction conditions per substrate pair, especially those with multiple reaction objectives. This limited scope is insufficient to benchmark HTE optimisation campaigns involving multiple 24/48/96-well plates. To address this limitation, we conducted benchmarks against emulated virtual datasets, following similar established practices[33,34] (Fig. 2a).

We trained ML regressors on reaction datasets from Torres et al.[14] (EDBO+), using ML predictions to emulate reaction outcomes for a broader range of conditions (e.g., temperature, concentration) than present in the original experimental training data (Fig. 2b). This emulation expands smaller experimental datasets to larger-scale virtual datasets more suitable for benchmarking HTE optimisation campaigns (Fig. 2c, d). Additionally, we incorporated similar virtual datasets from Olympus[33] in our benchmarking (Fig. 2a) (see Supplementary Information Section 1 for details on all benchmark datasets). To evaluate optimisation performance, we used the hypervolume metric[35] to quantify the quality of reaction conditions identified by the algorithms. The hypervolume calculates the volume of objective space (e.g., yield, selectivity) enclosed by the set of reaction conditions selected by our algorithm (Fig. 2e). The hypervolume considers both the convergence towards optimal reaction objectives and diversity, providing a comprehensive measure of optimisation performance. We compared the hypervolume (%) of the reaction conditions obtained by each algorithm to that of the best conditions in the original benchmark dataset, using the latter as a reference for the true optimal solutions.

Aligning with the standard number of reaction vials in solid-dispensing HTE workflows, we benchmarked our optimisation approaches using batch sizes of 24, 48, and 96 for 5 iterations, using Sobol sampling for initial batch selection in the first iteration. We compared the three acquisition functions (q-NEHVI, q-NParEgo, and TS-HVI) against a Sobol baseline. All acquisition functions showed comparable performance, outperforming the Sobol baseline on all datasets and demonstrating scalability to large batch sizes of 96. As convergence to optimal solutions was generally achieved on most benchmark datasets in 5 iterations with the lowest batch setting of batch size 24, we focused our comparative analysis on these results (Fig. 3a). Although the best acquisition function varied across each dataset, q-NEHVI consistently performed well on all benchmarks (see Supplementary Information Section 2 for statistical tests) and was thus selected for further analysis. We also compared our approach to other reaction optimisation software[14] (see Supplementary Information Section 2), demonstrating improved scalability to large batch sizes and search spaces. Our benchmark datasets included various reaction condition featurisation methods including Density Functional Theory (DFT), with high-dimensional reaction representations of up to 530 dimensions. While we have not explicitly tested upper computational limits of our approaches, we demonstrated robust scalability across dataset sizes, feature dimensions, and batch sizes examined in this study. All benchmark results can be found in the Supplementary Information.

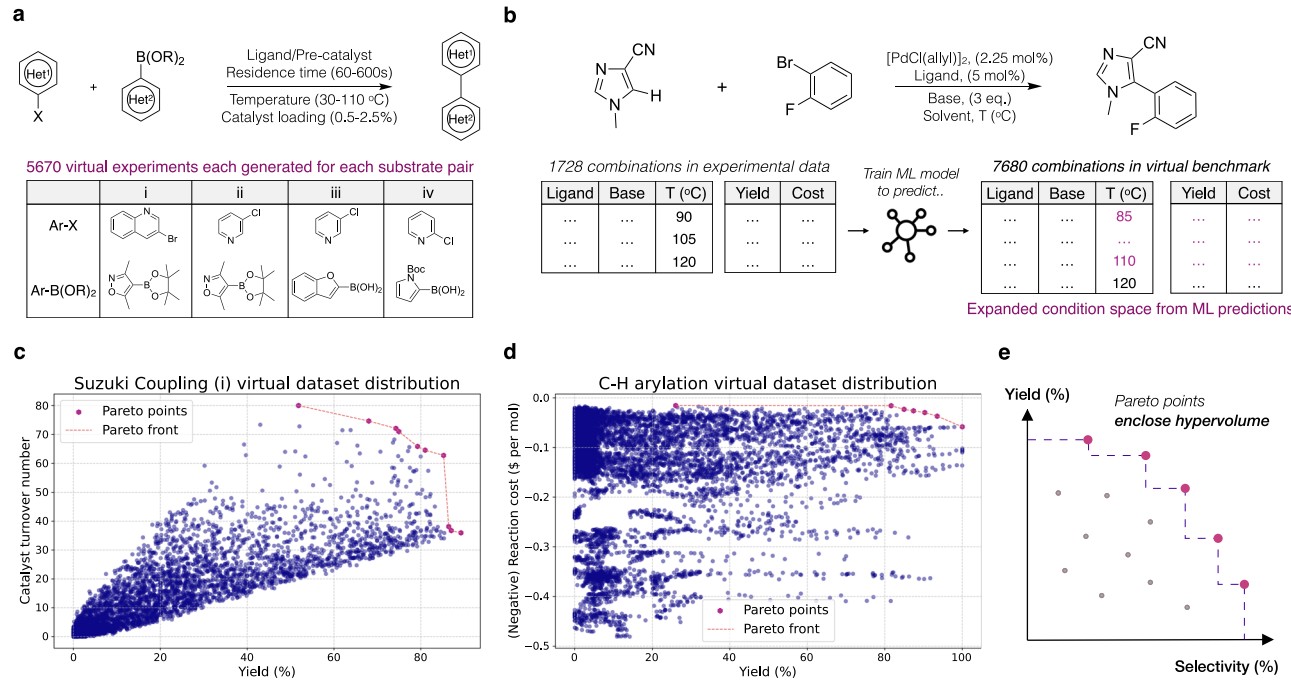

**Fig. 2 | Benchmarking techniques and metrics to assess optimisation performance. a** Four Suzuki coupling virtual datasets from Olympus[33], derived from experimental data, are used for benchmarking in this study. **b** The C-H arylation virtual dataset is generated by training a machine learning (ML) model on 1728 experimentally collected reactions from Torres et al.[14] (EDBO+), then predicting reaction outcomes for a larger range of reaction conditions and variables not present in the original training data. This creates a large-scale virtual dataset suitable for benchmarking high-throughput experimentation (HTE) optimisation campaigns (see Supplementary Information Section 1 and Methods). **c** Distribution of reaction objectives, yield (%) and catalyst turnover number, for the first Suzuki Coupling (i) virtual dataset from Olympus[33]. Pareto points represent the optimal multi-objective combinations. **d** Distribution of reaction objectives, yield (%) and reaction cost, for the C-H arylation virtual dataset generated in this study. **e** The hypervolume is used to assess the performance of optimisation algorithms. The hypervolume quantifies the volume enclosed by the optimal objective (e.g., yield-selectivity) combinations (Pareto points) identified by each algorithm. The hypervolume is used to compare algorithm performance against the best reaction conditions in the benchmark dataset, evaluating how effectively the best existing solutions are identified.

## Investigating noise robustness

Chemical reactions are stochastic in practice, resulting in yield variations even when reactions are repeated under identical conditions. Although HTE automation reduces this variability, some experimental noise, e.g., from dosing and stirring, is still expected. To assess the robustness of our optimisation workflow to such real-world variability, we simulated different noise levels by perturbing our model's input data with Gaussian noise of varying standard deviations (see Methods), comparing the performance of the q-NEHVI acquisition function under these conditions. Our results showed that while there is some reduction in performance due to input noise, our optimisation approach remains robust even at high noise levels, such as a noise standard deviation of 10 (Fig. 3b). In practice, we expect HTE automation equipment to exhibit lower levels of noise, and our approach demonstrates reliable performance under these conditions. The ability of our workflow to handle appreciable levels of noise highlights its potential for real-world applications, where experimental variability is unavoidable. All noisy benchmark results can be found in Supplementary Information Section 3.

## Equipment constraints on experimental batches

Our HTE platform utilises four heating wells for temperature control, restricting a single batch of experiments to a maximum of 4 unique reaction temperatures. To maintain practicality and minimise plate splitting across different heating wells, we restricted optimisation campaigns to 2 unique temperatures per batch. Such temperature constraints are also commonly observed in non-automated laboratory settings, for example, the finite number of heating plates in a fume hood. To implement this constraint into batch selection with our acquisition functions, we extended the naive and nested approaches from Vellanki et al.[36] to accommodate multiple unique allowed temperatures per batch and multi-objective acquisition functions. The naive approach uses standard Bayesian optimisation to select batch experiments until the number of unique temperatures meets the constraint (e.g., 2). The remaining experiments are then restricted to being selected from only those 2 unique temperatures. The nested approach first algorithmically selects the most promising temperatures (see Methods), then identifies optimal reaction conditions within the constraints of the selected temperatures. Both approaches can be extended to an arbitrary number of constrained temperatures per batch.

We evaluated these approaches in extreme cases, limiting batches to one or two unique temperatures (see Methods). Despite highly restrictive batch constraints, both approaches were able to identify high-performing experiments, yielding above 90 hypervolume (%) across all benchmarks (Fig. 3c). We observed similar performances between the two approaches for batch constraints of 2 unique temperatures. However, when constrained to 1 unique temperature, the naive approach consistently underperformed compared to the nested approach (see Supplementary Information Section 4 for statistical tests). Hence, for the experimental deployment of our workflow, we proceeded with the nested approach for handling batch constraints, combined with the q-NEHVI acquisition function. We note that while these two strategies were observed to perform the best overall across the benchmark datasets evaluated in this study, the optimal algorithm may depend on the specific dataset considered.

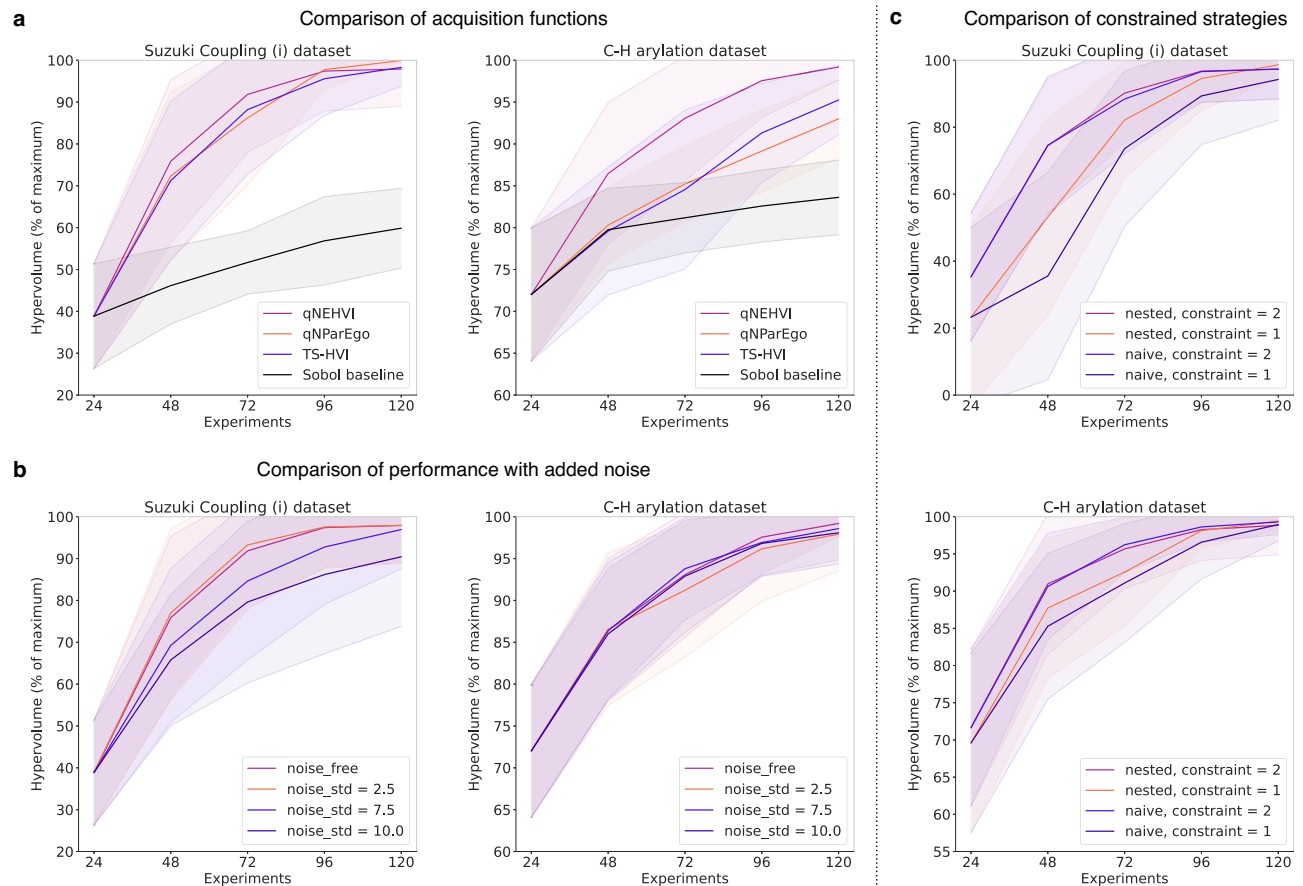

**Fig. 3 | Machine learning (ML) optimisation performance on the C-H arylation and Suzuki Coupling (i) virtual benchmark datasets.** All displayed benchmark results used a batch size of 24 for 5 iterations using the hypervolume (%) metric to assess reaction conditions identified by the optimisation algorithms (q-NEHVI[32], q-NParEgo[32], and TS-HVI[30]) compared to the best conditions in the ground truth dataset (see Supplementary Information Section 2 for benchmarks on all datasets and batch sizes 48/96). All optimisations were repeated across 20 different random seeds, plotting the mean hypervolume (%), with the shaded area corresponding to ±1 standard deviation. **a** Comparison of different scalable multi-objective acquisition functions against a quasi-random Sobol baseline. **b** Optimisation performance of the q-NEHVI acquisition function with varying magnitudes of Gaussian noise added to model observations. **c** Optimisation performance of the q-NEHVI acquisition function with the nested and naive constrained strategies to accommodate constraints of 1 or 2 unique temperatures per batch.

## Application to nickel-catalysed Suzuki reactions

Inspired by the successes of non-platinum group metal (PGM) catalysis in the pharmaceutical industry[3] and our lab's application of nickel catalysis to an active pharmaceutical ingredient (API), we challenged our optimisation workflow with a Ni-catalysed Suzuki-Miyaura coupling. Suzuki couplings are fundamental to pharmaceutical development, ranking among the top five reactions performed by medicinal chemists and the second largest reaction class at Roche and AbbVie[26,37]. Although Pd-catalysed Suzuki reactions dominate the industrial landscape, palladium's increasing scarcity, high cost, and substantial environmental impact (3880 kg $CO_2$ equivalents per kg of palladium) present challenges for its continued use[4]. In contrast, nickel offers a more sustainable and cost-effective alternative, with only 6.5 kg $CO_2$ equivalents per kg and a 3000-fold lower cost than palladium on a molar basis. These benefits become even more pronounced in process chemistry, where reactions are conducted on multi-(hundred) kilogram scales[38]. However, Ni-catalysed Suzuki reactions are more challenging, prone to substrate inhibition, and generate more byproducts compared to their palladium counterparts. Consequently, the development of Ni-catalysed Suzuki reactions is important for the pharmaceutical industry to address the challenges associated with palladium catalysis.

We selected a challenging nickel-catalysed heterocyclic Suzuki reaction between substrates **1** and **2** as an experimental case study (Fig. 4a). We defined a search space comprising 50 monophosphine ligands, 4 nickel precatalysts, 4 bases, 10 solvents, 3 co-solvents, and 5 temperatures. After removing conditions where the reaction temperature exceeded the solvent boiling point, we obtained a final reaction condition search space of 88,000 combinations. Given the large number of categorical variables, one-hot-binary encoding (OHE) featurisation was impractical. Instead, we employed DFT descriptors to represent ligands and solvents. To efficiently incorporate these high-dimensional chemical features, we leveraged dimensionality reduction techniques to enable effective exploration of the reaction space (see Methods for full reaction condition parameterisation). All experimental details are fully specified in Supplementary Information Section 5.

As a baseline comparison to traditional approaches, we first attempted to optimise this Suzuki reaction using chemist-designed HTE batches. Two 96-well HTE plates were designed based on chemical intuition and literature precedent, with the chemist having full access to first-plate results before designing the second (Fig. 4b). These plates yielded only trace amounts of product, with the best reaction achieving ~5 area percent (AP) yield based on Liquid Chromatography-Mass Spectroscopy (LC-MS) analysis (see Methods). In such scenarios, where initial screening fails to identify productive conditions despite iterative exploration, reactions may be prematurely deemed infeasible.

Subsequently, without incorporating the previous data, we re-attempted this optimisation from scratch, applying our ML optimisation workflow to this Suzuki reaction within the chemist-defined

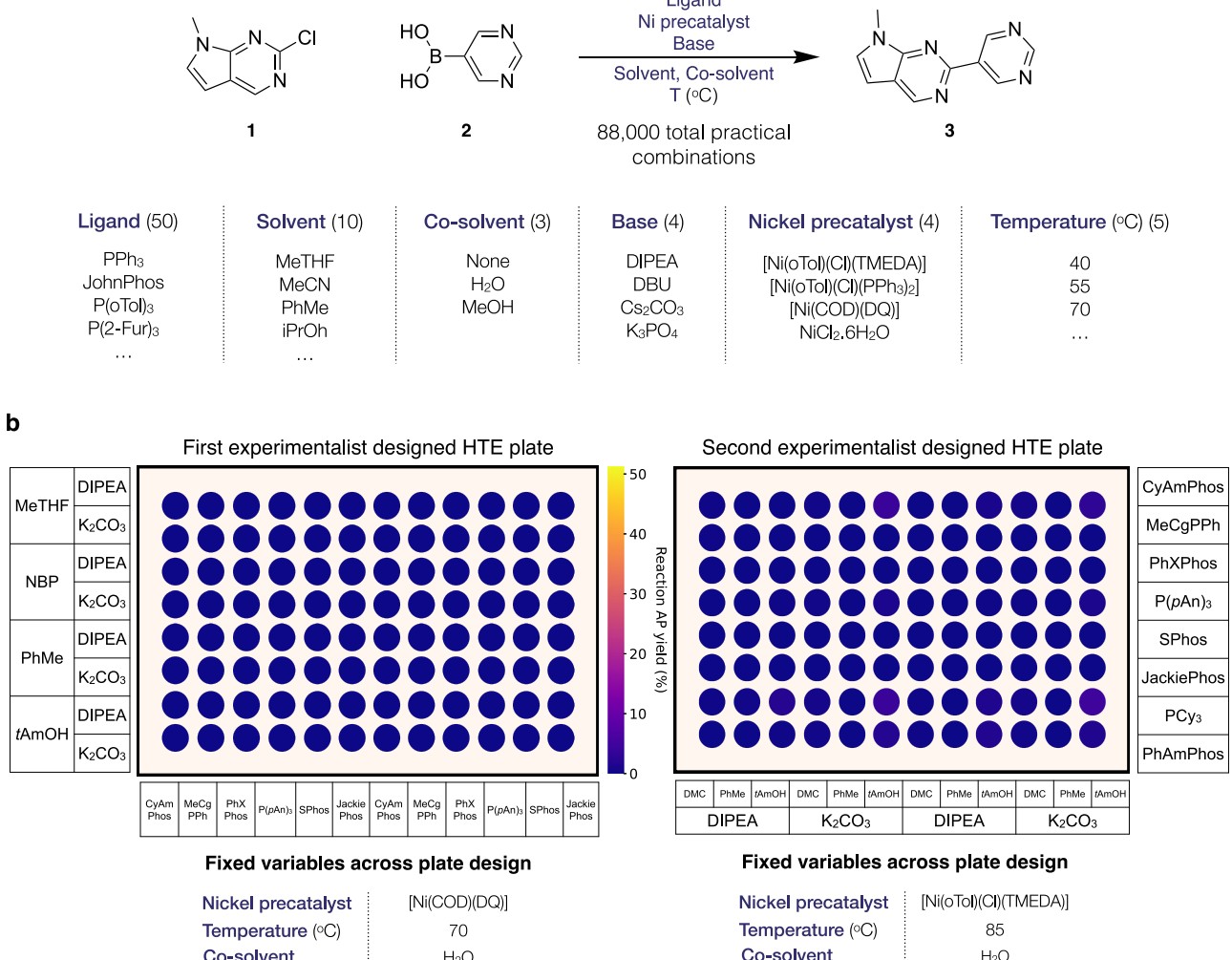

**Fig. 4 | Applying our machine learning (ML) optimisation workflow to a nickel-catalysed Suzuki reaction. a** The Ni-catalysed Suzuki reaction chosen for experimental application and the reaction parameters comprising the reaction condition search space defined by experimentalist knowledge. The multiplicative combination of these parameters initially yielded 120,000 possible reaction configurations. After removing configurations where reaction temperatures exceed the solvent boiling point, the final search space contained 88,000 possible reactions. The full search space is specified in Supplementary Information Section 5.2.1. **b** Initial optimisation results for the Ni-catalysed Suzuki reaction using traditional expert-driven high-throughput experimentation (HTE) plate design. Two 96-well HTE plates were executed, each containing 48 unique reactions run in duplicate to assess reproducibility (see Supplementary Information Section 5.2). The schematic shows the grid-like structured plate design and fixed variables, specifying the conditions evaluated in the two HTE plates.

search space of 88,000 reaction conditions. Our framework utilises systematic initial space exploration through Sobol sampling with ML-guided Bayesian optimisation in subsequent iterations. The first iteration, using Sobol sampling to select 96-well HTE batches, surpassed the results of the chemist-designed batches, identifying multiple reaction hits with up to 51 AP yield and 86 AP selectivity (Fig. 5a). Through ML-guided optimisation, the second and third iterations further improved on both AP yield and AP selectivity.

In the fourth iteration, with no improvement in maximum AP yield and selectivity beyond the values achieved in earlier iterations, we evaluated the optimisation trajectory to guide strategy selection for a planned final iteration. Based on emerging chemical insights from this transformation, and considering that further exploration was less critical in the final iteration, we assessed that the ML-suggested experiments for the fifth iteration were comparatively explorative. These suggestions proposed several reactions using PhMe solvent, which had been minimally explored until this point. At this stage, we deliberately steered the optimisation strategy towards exploitation, focusing on the thorough investigation of promising conditions identified thus far, at

the expense of further exploration. We adjusted the acquisition function to prioritise predicted performance over uncertainty exploration, selecting experiments with the highest predicted AP yield and AP selectivity values using Utopia point scalarisation (see Methods). This strategy in the final iteration identified multiple reactions with improved AP yield (up to 76%) with high AP selectivity (up to 92%) compared to previous iterations. These high-performing reactions, conducted at relatively lower temperatures beneficial for process scale-up, represent promising hits that would typically progress to fine-tuning of quantitative factors like catalyst loading and reagent equivalents in industrial process development. For comparison, we also performed the non-exploitative focused ML-suggested HTE plate for the final iteration. The superior performance of this adaptive strategy demonstrates how our ML-driven optimisation framework can effectively leverage accumulated campaign knowledge through expert-informed decisions in the final optimisation stages. (see Supplementary Information Section 5.2.6).

From experience, we expected initially that the choice of ligand would be the dominant factor influencing product formation, as is

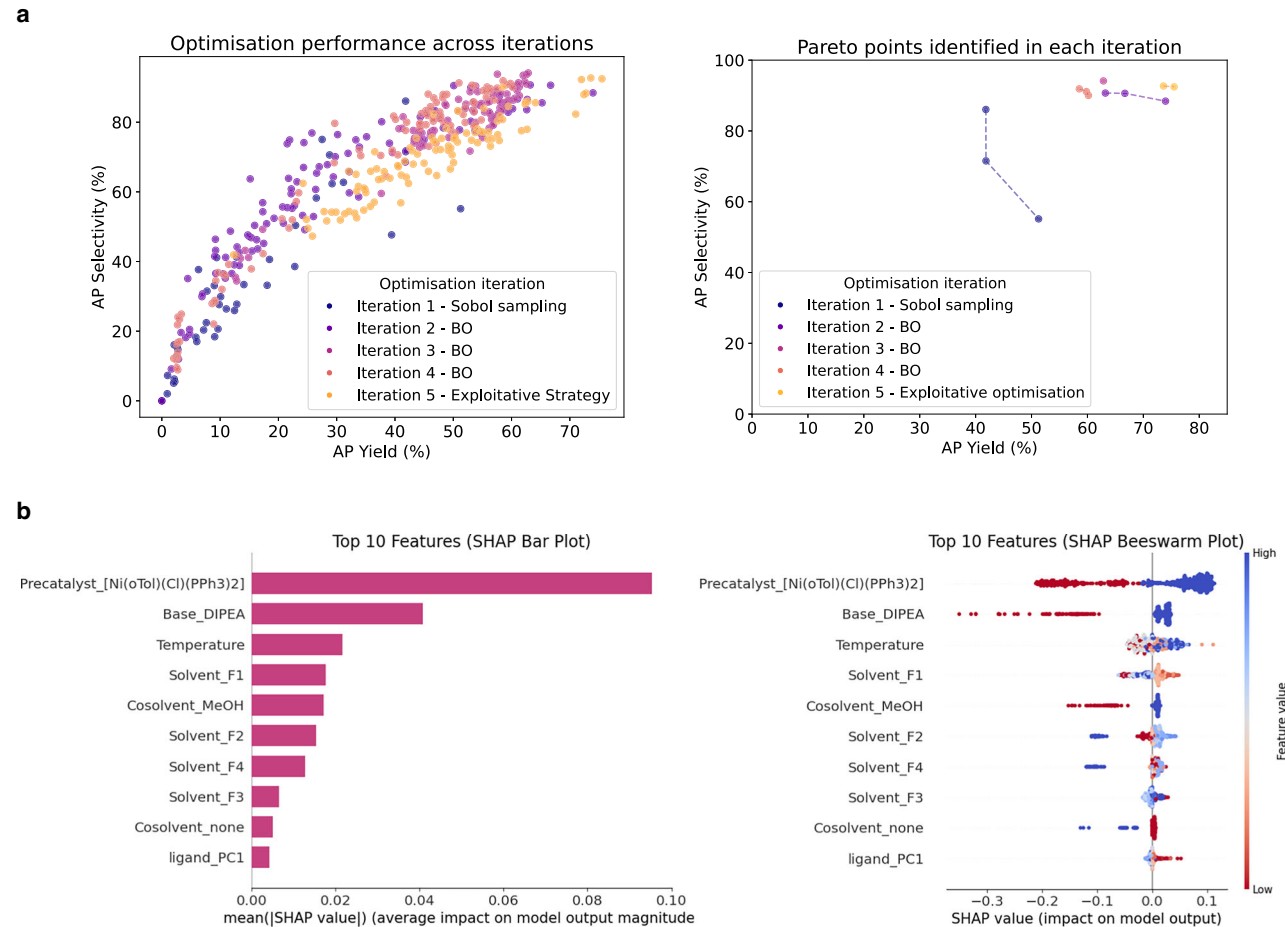

**Fig. 5 | Machine learning (ML) optimisation of the nickel-catalysed Suzuki reaction. a** The first scatter plot shows the area percent (AP) yield and AP selectivity of experiments selected by our ML Bayesian optimisation (BO) workflow at each iteration. The second plot shows the Pareto points, representing the best trade-offs between AP yield and selectivity, identified at each iteration. The Pareto points illustrate the optimal yield-selectivity combinations achieved. **b** Using all collected experimental data from the campaign (576 reactions), we trained an ML model to perform Shapley additive explanations (SHAP)[39] feature importance analysis. The SHAP values quantify the magnitude of each feature's contribution to the model's predicted AP yield, considering both positive and negative impacts to identify the most influential factors (see Methods for more details). The left panel shows the mean absolute SHAP values, while the right panel displays the individual data points that contribute to these means.

typical in cross-coupling reactions. However, observations from the campaign revealed that this Suzuki reaction was highly sensitive to multiple other parameters, particularly the choice of nickel precatalyst. [Ni(oTol)Cl(PPh$_3$)$_2$] was present in many high-performing reaction conditions, and successful reactions also depended strongly on the choice of base, solvent, co-solvent, and temperature. In contrast, ligand choice seemed to have comparatively minimal impact on reaction outcomes. A Shapley value analysis of all collected experimental data using SHAP (Shapley additive explanations)[39] corroborated these observations (see Methods), identifying the presence or absence of the [Ni(oTol)Cl(PPh$_3$)$_2$] precatalyst as the most important feature for AP yield, represented by the binary value of the one-hot encoded feature (Fig. 5b). Consistent with our observations, features representing bases, solvents, co-solvents, and temperature were also attributed substantially higher importance than those of the ligands. In fact, when we assessed the most promising conditions on a 500 mg scale, we observed essentially identical results with and without added ligand, indicating that [Ni(oTol)Cl(PPh$_3$)$_2$] catalyses the reaction. This unexpected chemical reactivity provides context for the minimal product formation observed in the two chemist-designed HTE plates using traditional approaches, which used the less performant [Ni(oTol)Cl(TMEDA)] and [Ni(COD)(DQ)] nickel precatalysts. Notably though, our ML workflow had identified productive conditions (>40 AP yield) for these same precatalysts, highlighting how optimal condition combinations beyond

precatalyst selection still influence reaction success. In large multi-dimensional reaction spaces with multiple parameters, as in this case study, systematic exploration and identification of optimal conditions becomes particularly challenging. In such complex landscapes, the presence of reactivity cliffs renders the reaction space challenging for experimentalists to navigate with traditional approaches and ensure comprehensive exploration[40]. Consequently, this could lead to premature conclusions about reaction viability. Our optimisation workflow provides chemists with a valuable tool to efficiently navigate and map these challenging and intricate spaces, potentially uncovering unexpected reactivity patterns missed by traditional approaches.

**Application to active pharmaceutical ingredient (API) synthesis**
Following successful experimental application of our ML-driven optimisation workflow, we demonstrated its practical utility in pharmaceutical process development through the optimisation of two active pharmaceutical ingredient (API) syntheses. We selected two case studies that represent both emerging and established high-value transformations in pharmaceutical synthesis: a Ni-catalysed Suzuki coupling and a Pd-catalysed Buchwald–Hartwig coupling. These reactions present distinct optimisation challenges which both deliver impact in pharmaceutical manufacturing.

Targeting a second Ni-catalysed Suzuki reaction allowed us to actively develop expertise and build further understanding of this

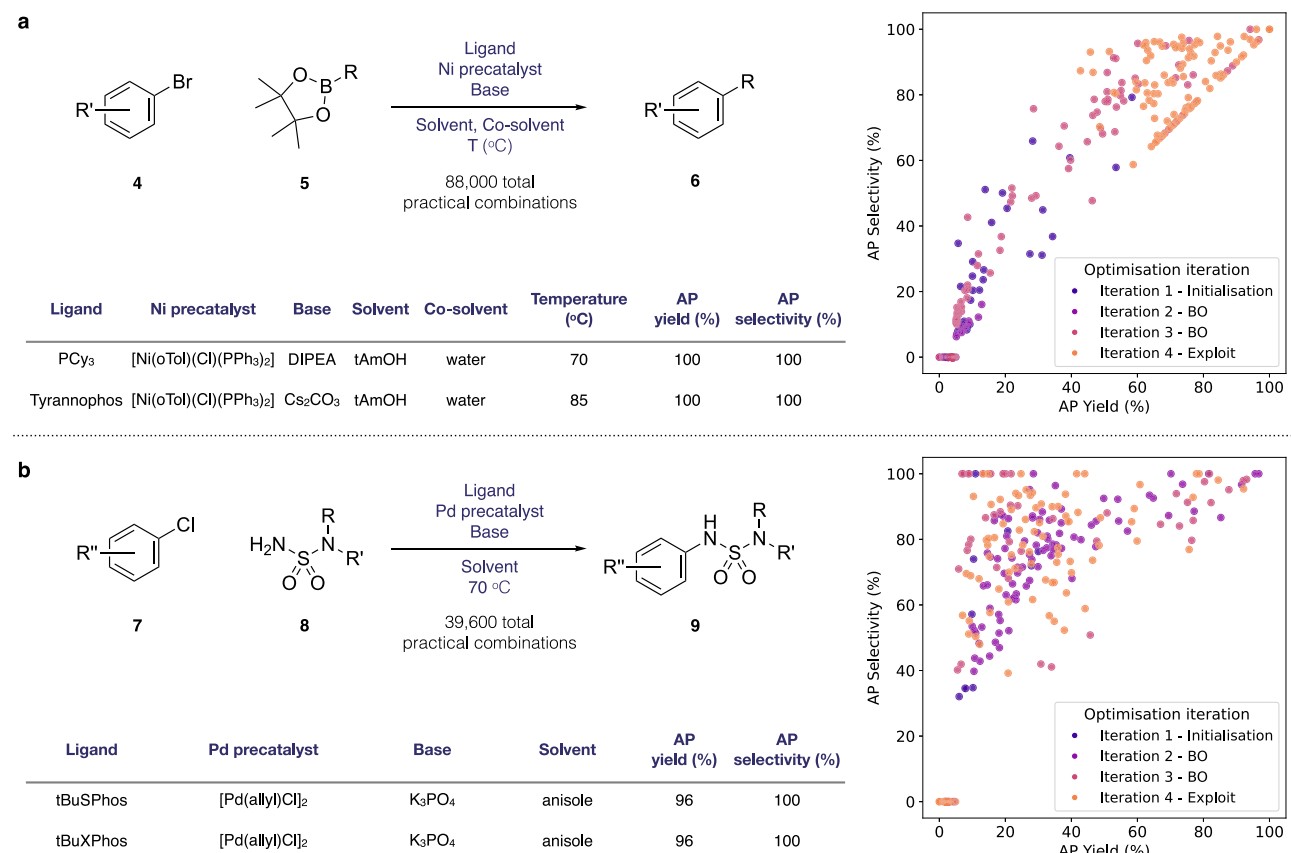

**Fig. 6 | Application of our machine learning (ML) optimisation framework to pharmaceutical process development with two active pharmaceutical ingredient (API) synthesis case studies.** The scatter plots show the area percent (AP) yield and AP selectivity of experiments selected by our ML Bayesian optimisation (BO) workflow at each iteration for each campaign. Select high-performing reaction conditions for each case study are presented in the tables. **a** Nickel-catalysed Suzuki reaction in an API synthesis. **b** Palladium-catalysed Buchwald–Hartwig coupling reaction in an API synthesis.

promising but less explored transformation. As highlighted previously, nickel catalysis offers compelling cost and sustainability advantages compared to palladium[4,41]. The greater mechanistic complexity and tendency for byproduct formation in nickel catalysis renders optimisation particularly challenging, providing another difficult test case for our ML workflow. Unlike nickel catalysis, Pd-catalysed Buchwald–Hartwig couplings represent one of the most important and widely employed transformations in pharmaceutical synthesis[42]. Their critical role is reflected in widespread industrial adoption, ranking among the top five reactions performed by medicinal chemists and representing one of the most frequently performed reaction classes both at Pfizer and in our HTE laboratories[43]. Despite this extensive precedent, process optimisation remains challenging due to strict pharmaceutical purity and yield requirements to offset the high cost of palladium.

By utilising our ML protocol with HTE batches of 96 reactions, we could efficiently evaluate diverse conditions in parallel for both the widely adopted Pd-catalysed Buchwald–Hartwig coupling and the emerging Ni-catalysed Suzuki reaction. This approach enabled us to systematically assess the optimisation capabilities of our ML-driven optimisation framework across reaction spaces with varying levels of established precedent and understanding.

**API case study 1: Nickel-catalysed Suzuki reaction.** For the Ni-catalysed API Suzuki coupling, we leveraged our understanding from the previous case study, employing the same reaction condition search space that proved successful in the previous campaign: 50 monophosphine ligands, 4 nickel precatalysts, 4 bases, 10 solvents, 3 co-solvents, and 5 temperatures, resulting in 88,000 possible reaction

combinations after accounting for solvent boiling point constraints. As before, we applied our workflow to this search space without prior experimental data.

Initial Sobol sampling in the first HTE batch of 96 experiments identified several reaction hits, achieving AP yields above 50%. While the same high-performing Ni-precatalyst from our previous study showed good activity, there was no clear discernible trend as to which factor most influenced the reaction outcome. In the second iteration, our ML protocol explored lower temperature regimes (40 °C/55 °C), allowing the evaluation of lower boiling point solvents such as MeOH. However, reaction activity was significantly diminished, with AP yields not exceeding 12%, suggesting insufficient reactivity of our reaction system at lower temperatures within our defined search space. Learning from experimental observations, our ML protocol re-focused on higher temperature regimes (70 °C/85 °C) in the third iteration and further explored new combinations of ligands, solvents, and bases. Our ML HTE batch design proved successful, identifying multiple conditions achieving >95 AP yield and up to >99 AP selectivity (Fig. 6a). tAmOH and iPrOH emerged as optimal solvents, while PCy₃ and Tyrannophos were identified as the most effective ligands.

With multiple high-performing conditions identified after just three iterations of our ML optimisation, we sought to exploit these promising regions of chemical space, deploying our ML framework's exploitative search for the fourth iteration. This focused optimisation strategy revealed additional reaction conditions achieving >99 AP yield and selectivity, with many reaction conditions above 95 AP yield and selectivity. Having identified multiple optimal conditions that exceeded our optimisation targets in just four batch iterations (~4 weeks of experimental cycle time), we concluded the optimisation

campaign. Insights from this optimisation campaign revealed interesting differences from our previous Ni-catalysed case study. While specific Ni-precatalyst choice remained important for identifying the top conditions, ligand selection emerged as a more critical factor, with PCy$_3$ and Tyrannophos consistently delivering optimal results across iterations 3 and 4.

The rapid identification of optimal conditions further validates our ML workflow's capability to systematically explore and map productive regions of chemical space for novel transformations without prior data, driving the development and uptake of these emerging, high-value transformations in pharmaceutical synthesis where domain knowledge and literature precedent are lacking.

**API case study 2: Palladium-catalysed Buchwald–Hartwig reaction.**
For the Pd-catalysed Buchwald–Hartwig API coupling, the development of an optimal process was already ongoing. While the existing process achieved a high yield, it suffered from significant formation of an impurity that proved very challenging to purge, a critical consideration given the stringent regulatory requirements for API purity in pharmaceutical process development. Given the prior success of our ML-driven optimisation framework, we hypothesized that our approach could identify improved process conditions. Using our domain expertise, we designed a comprehensive reaction space aiming to broadly search reaction conditions, encompassing 80 ligands, 3 palladium precatalysts, 11 base systems, and 15 solvents, resulting in 39,600 possible reaction condition combinations (see Methods). Without incorporating prior experimental data, we applied our ML-driven optimisation workflow to this coupling reaction.

Our ML-driven optimisation workflow rapidly identified promising conditions, with multiple reactions achieving >90 AP yield within two HTE batch iterations (Fig. 6b). Notably, several reaction conditions demonstrated >99 AP selectivity, forming only minimal amounts of the problematic impurity which compromised the prior existing process. Continuing with our workflow, we conducted two additional optimisation rounds, concluding with an exploitation-focused iteration to thoroughly investigate these promising leads.

Chemical analysis of the high-performing conditions revealed that ligands with sterically hindered phosphorus centres with tBu substituents, particularly TrixiePhos, tBuSPhos, and tBuXPhos, delivered superior performance. The [Pd(allyl)Cl]$_2$ and [Pd$_2$(dba)$_3$] Pd precatalyst combined with K$_3$PO$_4$ emerged as optimal choices across multiple conditions. Notably, high activity was maintained across many diverse solvents encompassing protic, aprotic, polar, and non-polar solvents such as toluene, 2-methyl-2-butanol, and anisole. Execution of the high-yielding reaction conditions at scale with multiple solvents culminated in improved process conditions. Scaled-up reactions at gram scale with tBuSPhos, K$_3$PO$_4$, and [Pd(allyl)Cl]$_2$ in anisole demonstrated superior performance compared to the previous process, reducing the problematic impurity from 3.8% to <0.5% while achieving >99% conversion (see Supplementary Information Section 5.4.3).

Importantly, this optimisation campaign was completed in just 4 weeks, compared to the 6 months required for the original process development. This dramatic acceleration in the optimisation timeline, even for a well-understood transformation, demonstrates how our ML-driven HTE optimisation framework can substantially expedite pharmaceutical process development towards drug synthesis. These results highlight that even for established reaction classes where considerable chemistry knowledge exists, our systematic ML-guided optimisation can reveal valuable opportunities for process improvement.

## Discussion
Our study demonstrates the efficacy of our ML-driven optimisation framework in an automated, high-throughput setting. Optimisation initialisation with algorithmic quasi-random sampling ensures comprehensive coverage of large reaction spaces to identify promising

local optima and reactivity crevices. Subsequent ML-driven optimisation iteratively refines reaction objectives, with exploitation-focused search in the final round to leverage all accumulated knowledge to maximise final reaction outcomes. We successfully validated our ML-driven optimisation workflow on a challenging Ni-catalysed Suzuki reaction in a 96-well HTE campaign, demonstrating advantages over traditional, purely experimentalist-guided methods. Further validation through pharmaceutical process development demonstrated our workflow's broader utility: we rapidly optimised two API syntheses, identifying multiple reaction conditions with >95 AP yield and selectivity for both a Ni-catalysed Suzuki coupling and a Pd-catalysed Buchwald–Hartwig reaction. In one case, our approach identified improved process conditions at scale in 4 weeks compared to a previous 6-month development campaign. By integrating machine learning strategies with highly parallel automation, our workflow demonstrates real-world applicability for accelerating reaction optimisation and is currently being implemented in several process chemistry projects at Roche. New values of parameters such as ligands and solvents could also be added to the search space as optimisation proceeds[14], and the workflow can be applied to other chemical reactions of interest, including those outside of HTE settings.

In this work, we only conducted the HTE optimisation campaign using a static search space across all iterations comprising 88,000 possible reaction combinations, defined by the experimentalist prior to initialisation. Future applications could benefit from dynamic, experimentalist-guided modifications to the search space during iterations, combining observed data with expert knowledge. Moreover, integrating more systematic experimentalist input could guide the desired balance between exploration and exploitation at various stages of the campaign, better aligning the process to specific practical requirements. The suggested improvements could be particularly effective when combined with explainable AI and visualisation methods, enhancing interpretability and decision-making. Furthermore, connecting explainable SHAP analysis with reaction-specific chemical descriptors could bridge ML predictions and fundamental chemical principles, enabling deeper mechanistic understanding. By providing chemists with a clearer understanding of the ML-guided optimisation process and the rationale behind its suggestions, we can foster greater trust and adoption of ML optimisation approaches within the chemical community. We envision that such collaborative approaches between ML algorithms and human experts could substantially enhance the efficiency and effectiveness of reaction optimisation in academia and in pharmaceutical development.

## Methods
### Emulated benchmark datasets
We used ML regressors trained on experimental data to generate expanded reaction condition spaces by predicting reaction outcomes for a larger set of experiments that, while not present in the original training data, fall primarily within the domain of the training set. We refer to these ML regressors as emulators. We used emulators from the Olympus[33] benchmarking framework (*suzuki_i* to *suzuki_iv*) to generate the Suzuki Coupling (i to iv) virtual benchmark datasets. Similarly, we also trained a Multi-layer Perceptron (MLP) using TensorFlow (2.13.1)[44] on experimental data from Torres et al.[14] (EDBO+) to generate the C-H arylation virtual benchmark dataset. A detailed description of all (emulated) virtual benchmark datasets is included in Supplementary Information Section 1 and the data accompanying this paper. Additionally, we have also created expanded virtual benchmarks from the HTE experiments we have conducted in this study for further analysis (see Supplementary Information Section 2, Figs. 8 and 9).

### Computational implementation
Bayesian optimisation is an iterative approach for identifying the local optima of 'black-box' functions, where the functional form is unknown

or cannot be expressed analytically. It relies on constructing a probabilistic surrogate model, usually a Gaussian Process (GP)[29], to approximate the unknown function. An acquisition function, using the GP model's predicted mean and uncertainty, selects the next experiments to evaluate based on previously observed data. The acquisition function balances exploration and exploitation, simultaneously enabling efficient exploration of uncertain regions in the parameter space whilst exploiting promising regions, allowing the algorithm to converge towards local optima. Frazier[45] provides additional background on a more comprehensive mathematical treatment of Bayesian optimisation. In this work, we used the multi-objective acquisition functions: q-NParEgo[32], Thompson sampling with hypervolume improvement (TS-HVI)[30], and q-Noisy Expected Hypervolume Improvement (q-NEHVI)[32] in the development of our optimisation workflow.

Multi-objective optimisation problems can be approached through scalarisation, which combines multiple objectives into a single objective, or through direct assessment of Pareto front improvements using hypervolume metrics. q-NParEgo is a noise-robust extension of the ParEGO algorithm allowing for batched acquisition function evaluations[31,32,46]. q-NParEgo models multiple objectives as independent GPs, using augmented Chebyshev scalarisation to reduce them into a single composite objective, enabling scalable optimisation[31,46]. For each batch candidate, q-NParEgo uses a different weight vector to scalarise objectives, randomly selected from a uniform distribution, allowing exploration of diverse trade-offs between objectives along the Pareto front[46]. By optimising the composite objective with Noisy Expected Improvement, which integrates under possible realisations of the true function given noisy observations, evaluations of q-NParEgo are more robust to observation noise[47].

Beyond scalarisation, candidate points can be evaluated through hypervolume improvement (HVI), which measures their contribution to expanding the hypervolume beyond the current Pareto front[35]. The expected hypervolume improvement (EHVI), defined as the expectation of HVI over the posterior distribution, provides a principled way to evaluate candidate points. Popular acquisition functions such as q-EHVI[31], used in prior chemical reaction optimisation applications[14], evaluate the joint hypervolume improvement of batch candidates using the inclusion-exclusion principle (IEP). However, q-EHVI assumes no observation noise, and IEP methods for computing joint hypervolume improvements scale exponentially with batch size and lead to memory overflow, causing prohibitive computational costs[30,32]. We empirically confirm these computational limitations in reaction optimisation frameworks like EDBO+ (see Supplementary Information Section 2). Thus, we sought to utilise hypervolume-based acquisition functions that offer noise robustness and scalability with batch size in our framework.

TS-HVI, Thompson sampling with hypervolume improvement, is a scalable approximation of q-EHVI[30]. For a batch size q, TS-HVI draws q posterior samples from the GP, and for each singular sampled function realisation, selects a candidate point that maximizes the hypervolume improvement over the Pareto front to comprise q candidate points. This single-sample approximation provides a computationally efficient alternative to q-EHVI, which requires computing the full expectation over the posterior distribution[31].

q-NEHVI assumes noisy observations, and that the observed Pareto front from experimental observations is a noisy realisation of the true Pareto front[32]. q-NEHVI handles noise by integrating the EHVI over possible true function realisations given the noisy observations. q-EHVI computes EHVI relative to possible Pareto frontier realisations, maintaining Bayes-optimality under noisy observation environments. q-NEHVI also uses cached box decompositions (CBD) instead of the inclusion-exclusion principle (IEP) used in q-EHVI[31], reducing computational complexity from exponential to polynomial in batch size. This improvement prevents GPU memory overflow and significantly reduces computation time, enabling larger batch sizes that are infeasible

with IEP-based approaches. We used PyTorch (2.0.1)[48], GPyTorch (1.6.0)[49], and BoTorch (0.6.0)[50] to build, initialise, test, benchmark, and apply all of our GP surrogate models and acquisition functions.

Initialisation of training data for our ML workflow in all cases was generated using Sobol sampling methods with PyTorch[48], employing low-discrepancy Sobol sequences to obtain quasi-random points diversely covering a unit hypercube, providing superior distribution and space-filling properties than standard random sampling[28]. We then map these sampled points to a discrete multi-dimensional reaction search space to obtain the initial reaction condition suggestions. Similar methods like centroidal Voronoi tessellation (CVT) and Latin hypercube sampling (LHS) have shown superior optimisation performance compared to random initialisation[14]. Training inputs and targets for all GPs are normalised and standardised, respectively, according to the specifications in BoTorch[50]. As we aimed to focus the benchmarks on comparing differences between acquisition functions, we used a GP with general purpose kernel hyperparameters adapted from EDBO+[14] for all benchmarks. All acquisition functions were implemented using BoTorch[50]. We implemented KeOps (1.5)[51] and fast variance estimates from GPyTorch[49] to enable memory-efficient computations. We used PyTorch Lightning (2.1.2)[52] to set random seeds, running 20 repeats from seed 1–20 for each benchmark result. All computations were run on a workstation with an AMD Ryzen 9 5900X 12-Core CPU and an RTX 3090 (24GB) GPU.

Comparisons against EDBO+[14] were implemented according to instructions in the code accompanying its publication, using the same scripts but replacing the Pd-catalysed C-H arylation data set in the EDBO+ directory with the expanded C-H arylation virtual benchmark dataset generated in this work. To ensure fair comparison, EDBO+ was tested with both CPU and GPU (see Supplementary Information Section 2). For noisy benchmarks in Section "Investigating noise robustness", we perturbed objective values only for yield and catalyst turnover number for all datasets, excluding the noiseless reaction costs in the C-H arylation virtual dataset. The noisy values were clamped at 0% and 100% for yield, and for turnover in the Suzuki Coupling virtual datasets, at 0 and the maximum observed values. The nested acquisition function described in Section "Equipment constraints on experimental batches" first selected promising temperatures by ranking the temperatures with the highest average q-NEHVI acquisition function value, followed by standard acquisition function evaluation on the search space restricted to the obtained top performing temperatures. For the benchmarks constrained to 1 and 2 unique temperatures per batch, model initialisation data was constructed by randomly selecting 1, or 2 unique temperatures according to constraints. Then, Sobol samples were drawn from experimental conditions restricted to those temperatures to initialise the optimisation process. All Wilcoxon statistical significance tests included in the Supplementary Information were implemented using SciPy (1.10.1)[53].

Exploitative greedy reaction optimisation strategies in a single-objective case select experiments based solely on only the highest predicted mean yield %[20], neglecting any GP model uncertainty and hence exploration. For our implementation of multi-objective exploitative optimisation, we used Utopia point scalarisation to combine AP yield and AP selectivity into a single value. Utopia point scalarisation measures the Euclidean distance of all AP yield and AP selectivity data values from a hypothetical ideal Utopia point, set in this case to 110 AP yield and AP selectivity, and has shown effectiveness in prior Bayesian optimisation applications[23]. This scalarisation allowed us to rank and select the most promising reactions considering both objectives simultaneously. Analogous to the single-objective case, we selected experiments with the predicted mean AP yield and AP selectivity values closest to the Utopia point. Consistent with an exploitative greedy approach, we selected the two temperatures for this experimental batch using the experiments with the closest predicted Utopia point distances.

## Experimental application

The reaction condition space for both Ni-catalysed Suzuki couplings comprised 50 monophosphine ligands, 4 nickel precatalysts, 4 bases, 10 solvents, 3 co-solvents, and 5 temperatures to give a total of 120,000 combinatorial reaction configurations (see Supplementary Information Section 5.2.1 for the full list). After removing conditions where the reaction temperature exceeded the solvent boiling point, we obtained a reaction condition search space of 88,000 configurations. Given the large number of categorical variables, which would require 71 one-hot-encoded (OHE) features to describe, we parameterised monophosphine ligands and solvents using quantum mechanical descriptors, which have shown good performance in prior optimisation studies[14,24,54,55], to provide a more informative representation.

We used 190 monophosphine ligand DFT descriptors from the Kraken featurisation workflow[56]. As high-dimensional representations require many data points for GP surrogate models to learn meaningful trends[30], we applied principal component analysis (PCA) using scikit-learn (1.3.2)[57] to reduce the dimensionality of the ligand descriptor space to 37 principal components based on a 99% explained variance threshold, labelled *ligand_PC*1 to *ligand_PC*37. This allowed us to capture essential chemical information from ligands whilst maintaining a computationally tractable optimisation problem. Similarly, solvent descriptors were obtained from Moity et al.[58] with parameterisation using COSMOtherm and represented with 4 DFT-based descriptors labelled *Solvent_F*1 to *Solvent_F*4. The rest of the categorical variables (co-solvents, bases, and precatalysts) were featurised using OHE, with temperature remaining numerical.

For the Pd-catalysed Buchwald–Hartwig reaction case study, the reaction condition space comprised 80 monophosphine ligands, 3 palladium precatalysts, 11 base systems, and 15 solvents to give a total of 39,600 possible reaction condition configurations (see Supplementary Information Section 5.4.1 for full list). Similarly, we featurised monophosphine ligands using physical chemistry descriptors from the Kraken workflow[56], applying PCA to narrow down the 190 ligand descriptors from Kraken to 25 principal components based on a 95% explained variance threshold, labelled *ligand_PC*1 to *ligand_PC*25. Solvents were once again featurised using COSMOtherm, and the rest of the categorical variables (bases and precatalysts) featurised using OHE. The resulting encoded reaction feature spaces are provided in the data section.

Our HTE platform from Unchained Labs uses four distinct heating wells for reaction execution. We used two heating plates for each batch of HTE experiments in our ML optimisation workflow, constrained to two unique temperatures per batch. The experiments used to initialise the ML experimental workflow were selected using quasi-random Sobol sampling, restricting initial reaction temperatures to 70 and 100 degrees Celsius (see Supplementary Information Section 5 for experimental procedures). All HTE reactions were evaluated using area percent (AP) yield. These yields, derived from Liquid Chromatography (LC), are uncorrected from differences in LC response factors between the Suzuki coupling product and the limited starting material. While approximate, AP yields provide a useful measure for elucidating reactivity trends and comparing the performance of different reaction conditions[38]. Detailed specifications of the HTE platform and comprehensive descriptions of analytical methods are provided in the Supplementary Information Section 5.1.

## Experimental data analysis and visualisation

To support empirical observations from the collected HTE experimental data and further elucidate underlying chemical relationships governing the nickel-catalysed Suzuki reaction (Section "Application to nickel-catalysed Suzuki reactions"), we employed several analysis and visualisation methods. First, we generated box plots comparing the average experimental AP yield value of all ligands, solvents, bases, precatalysts, co-solvents, and temperature to the overall average AP yield of all reactions. These plots are included in Supplementary Information Section 5. To gain deeper insights into feature importance, we trained a Random Forest surrogate model on the collected HTE data to approximate the AP yield function. We then applied SHAP (0.44.1)[39] analysis to obtain feature importances for features of each experimental parameter. SHAP uses cooperative game theory concepts to quantify the magnitude of each feature's contribution to the model's prediction, assigning each feature a SHAP value that represents its impact on model output. Beeswarm and bar plots of the SHAP results were generated using the default settings of the SHAP[39] package. For a more comprehensive explanation of the SHAP methodology and its implementation, we refer the reader to the SHAP package documentation, which provides more detailed information on the calculation and interpretation of SHAP values. All plots in this study were generated with Matplotlib (3.7.4)[59].

## Reporting summary

Further information on research design is available in the Nature Portfolio Reporting Summary linked to this article.

## Data availability

All virtual and high-throughput experimentation (HTE) experimental data and reaction condition spaces generated in this study are included in the manuscript, Supplementary Information, and on the accompanying public GitHub repository. Due to commercial and intellectual property (IP) considerations, the API structures in the API experimental case studies are partially obscured and are not publicly available. Confidential access can be obtained by entering a legal agreement with F. Hoffmann-La Roche and requesting permission under reference to Blue Sheets request 00067826. Source data are provided with this paper.

## Code availability

The Python code used in this study is made available in a public GitHub repository under the MIT open source license: github.com/schwallergroup/minerva[60].

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

## Acknowledgements

We thank T. Vögtlin, M. Müller, T. Chi, and J. Krizic from the High-Output Reaction Screening System (HORSS) team at F. Hoffmann-La Roche Ltd. for experimental and analytical support. We thank S. Fantasia and E. Trachsel from the Catalysis & Flow Reactions team at F. Hoffmann-La Roche Ltd. for experimental support and useful discussions. J.W.S., R.P. Burwood, K.P., and R. Bigler thank Roche and its Technology Innovation and Science initiative for generous financial support. S.L.C. acknowledges support from the Helmholtz Association of German Research Centres. P.S. acknowledges support from NCCR Catalysis (grant no. 225147), a National Centre of Competence in Research funded by the Swiss National Science Foundation. We are grateful to S. Chen, M. Adachi, Z. Jončev, J.J. Dotson, R.C. Walroth, and K.A. Mack for useful discussions.

## Author contributions

J.W.S. contributed to the conceptualisation, methodology, code development, analysis, visualisation, and writing of the manuscript. S.L.C. contributed to methodology, code development, and writing of the manuscript. R.P. Burwood and K.P. contributed to conceptualisation, supervision, and writing of the manuscript. R. Bigler contributed to conceptualisation, supervision, experimental results, and writing of the manuscript. P.S. contributed to conceptualisation, methodology, supervision, and writing of the manuscript.

## Competing interests

J.W.S., R.P. Burwood, K.P., and R. Bigler declare potential financial and non-financial conflicts of interest as full employees of F. Hoffmann-La Roche Ltd. The other authors declare no competing interests.
