## [Peer Review file · Nature Communications]

Highly Parallel Optimisation of Chemical Reactions through Automation and Machine Intelligence

Corresponding Author: Mr Joshua Sin

Version 0:

Reviewer comments:

Reviewer #1

(Remarks to the Author)

The manuscript "Highly Parallel Optimisation of Nickel-Catalysed Suzuki Reactions through Automation and Machine Intelligence" by Sin et. al., details using automated HTE experimentation guided by autonomous ML protocols to optimize a Ni-catalyzed Suzuki reaction. As expressed to the journal editors, as an expert in HTE chemistry, with no practical training in ML techniques, I can only weigh in on the experimental aspects of the work.

I have thoughts on several aspects of the experimental work:

1.) Reaction choice. The choice of a Ni-catalyzed Suzuki reaction for optimization has positive and negative aspects. On the positive side, it is a very important transformation, especially in Process Chemistry, where the use of well-established Pd-catalysis is problematic for reasons described in the text. While it is a high-value goal to pursue this reaction with autonomous optimization, I feel that it is a relatively niche reaction where the "expert-user" comparison is practically guaranteed to work, since most chemists have little practical experience to draw from. The rules of ligand/ base/ solvent choice for Ni reactions are far less generally understood than for the Pd-variant. Perhaps a description of the expert-users experience would be helpful here (i.e., they have significant experience optimizing Ni-catalyzed reactions). Even the literature does not have a strong representation of this type of reaction to draw from.

2.) HTE experimentation. I have several concerns with the experimental set-up as described in the supplementary information.

a. First, the experimentation is only very basically described. A more full description of the experimental set-up is required to understand the quality of the experiments. It appears that the experiments were set-up on an Unchained Labs robotics platform (from the text, not described in the SI). All solids were added to vials first, followed by liquid components. It would appear from the description that exceedingly small amounts of several components are added to the reactions (Ni pre-catalysts and ligands). In my experience, adding 0.2 mg of Ni pre-catalyst (3% loading) will have a very high error bar with a solid handling robot. As well, with each reaction having a unique solvent, there is time associated with dosing solvent at the end, which may lead to solvent evaporation and variable concentration. While these "errors-bars" might be acceptable in a "discovery phase" of optimization- i.e., finding a workable starting point, a high error bar in subsequent continuous optimization would lead to a high degree of variability, minimizing the ability to fine-tune conditions. Also, I feel that experimental sections of synthetic organic papers should be reproducible, so it is important to document how much of different reagents are added in reactions. Most robotics systems will provide a report on dosing, this should be included.

b. Second, as an experienced HTE experimentalist, I am concerned about the manner in which the experiments were conducted. In order to make the reaction components weighable (large enough quantities), they were conducted at 60 μmol scale, in 120 μL of a wide variety of solvents (non-polar to highly polar). Running at high concentration and with a variety of solvents will invariably lead to a high degree of reaction heterogeneity. The mixing approach is poorly described in the experimental section ("stirring disks"). Some validation (or a reference) where this method is effective at agitation at this scale and concentration would be important. In addition, the "mix it all together" approach is often a very poor way to form metal-ligand complexes. Catalyst pre-formation, and ideally, activation are tools that process chemists will invariably rely on to improve reaction performance. The issue with inadequate mixing and catalyst formation experiments is that the results will be enriched in the most robust experiments, rather than the reactions that have the best yield potential. Robustness is a critical aspect of process chemistry but can typically be built into process development once suitable conditions are identified. In this HTE optimization experimentation, I believe the conditions should not have this robustness bias built in.

c. Third, the analytical approach is odd to me. If the reaction are all giving the same product, then standard dilution of reactions should result in a product area count assessment, where the peak area is directly proportional to product formation. Given that then actually made and purified the product independently, it is possible to calculate the LC assay yield. Use of LC area percent is going to lead to lower quality data, as other UV-absorbing compounds (ligands for example) will lead to greater variability in the data. It should be possible to fix this retro-actively and report the data as actual yields, not LCAP.

d. Finally, I would like to see a scale-up of the best results identified.

3.) Comparison to expert guided work. One of the key experiments in this work is the comparison of autonomous optimization to expert guided experiments. As mentioned above, limited expertise with Ni chemistry is likely to reduce the "expert's" ability to come up with a good approach to solving this problem, relative to more well-used experiments. As well, the key finding, that Ni-pre-catalyst is the main determining factor, and that ligands are not important in the optimization, is the key insight that enabled reaction optimization. I believe that if the expert user was given this information, they could have readily optimized the reaction, especially with 4 additional rounds of HTE optimization. What this suggests is that the expert user comparison should really be accomplished after each step to do an adequate comparison, I actually think that the next 3 rounds of autonomous optimization, in which the algorithm chose to mostly run more exploratory experiments, rather than exploitative experiments (and the authors needed to change the protocol on the last step), suggests that the autonomous HTE experiments were not particularly well executed. In the end 79% yield after an initial ~50% yield in round 1, suggests that the additional 4 rounds of HTE were not particularly impactful. In the end, I don't find the comparison particularly compelling, but this system did come up with a reasonable solution, albeit with some human intervention, which is commendable.

4.) Punchline. I feel that papers that describe new tools, particularly in high-end journals, should have a strong punchline that clearly demonstrates the value of the work. In my assessment, and again, without being able to weigh in on the ML aspects of the work, the punchline in this story is fairly weak. The most impactful aspects of the work appear to me to be the broad coverage of conditions using HTE in the first step (Sobol sampling), which enabled the identification of Ni pre-catalyst as the main driver in reaction performance. The additional rounds of autonomous, ML-driven experimentation did not appear to add a large amount of value. With a workable starting point, a typical chemist should be able to improve a reaction in similar or better efficiency. In the end, the more important point may be that the autonomous system was able to solve a difficult chemistry problem, albeit with some human intervention. This is the likely outcome of "autonomous approaches", wherein people and computers will work together to solve problems. In the end, if their new autonomous optimization approach is truly enabling, they should be able to generate a more compelling story, in an efficient manner.

My opinion is that this work may have innovative aspects in terms of the ML optimization techniques used, but comes up quite short of the mark in terms of the chemistry documentation, execution and the punch-line. In the end, if their new autonomous optimization approach is truly enabling, they should be able to generate a more compelling story, in an efficient manner and with better documentation.

(Remarks on code availability)

Reviewer #2

(Remarks to the Author)

This article describes the development of a machine learning (ML) optimization framework for batched multi-objective reaction optimization, and overall strategy is well organized. The optimization campaign identified reaction conditions with a 76% yield and 92% selectivity, which is a substantial improvement over the results obtained from traditional expert-designed high-throughput experimentation (HTE) plates that failed to yield successful reaction conditions. This showcased the practical value and effectiveness of the proposed approach in solving real-world reaction optimization problems. The robustness of the framework to reaction noise and its ability to handle batch constraints encountered in real-world laboratories are also important contributions. This makes the approach more applicable in practical settings where such factors are inevitable.

In general, the optimization framework developed here is innovative and practical, with good example described which further demonstrate some industrial applicability of this method. However, here are some specific comments for this article, which are suggested to be taken care of by the authors before this article get published:

- The authors need to more clearly demonstrate the similarities and differences between the newly developed framework and existing approaches, i.e., q-NParEgo, q-NEHVI, TS-HVI. What's the main improvements and advantages?
- Although significant progress has been made in optimizing this challenging nickel-catalyzed Suzuki reaction through Automation and Machine Intelligence, it is notable that the optimal results obtained from the machine learning approach have not been transferred to benchtop for gram-scale validations and further scale-up, leading to some uncertainties in the practical value of the optimization results generated from the framework.
- The ML process started with Sobol sampling, aiming to maximize the coverage of the reaction space. Subsequent iterations involved model-guided Bayesian optimization, and an exploitative optimization strategy was implemented in the final iteration. Would 5 rounds of iterations sufficient for a standardized coupling reactions? If more rounds of iteration optimizations performed for the Suzuki case, would you expect higher yield of the desired product?
- Although SHAP and other methods are used for feature importance analysis, the decision-making process of the machine learning model may still lack sufficient intuitiveness for process chemists in industry. How can the predictions of the machine learning model be further integrated with chemical intuition and domain knowledge? For example, could more explicit chemical explanations be provided to clarify why certain features (such as the nickel pre-catalyst) have a significant impact in the model and how these features are related to known chemical principles?

- The article demonstrates the successful application in the nickel-catalyzed Suzuki reaction. However, what is the generality of this optimization framework in other types of chemical reactions? Have more extensive tests been conducted or is there a theoretical analysis to support its applicability in different reaction systems? Different reaction types may have diverse reaction kinetics, substrate characteristics, and optimization requirements. The flexibility and effectiveness of the framework in the face of these variations need to be further explored.
- Is it possible for the authors to demonstrate a real industrial case of applying such learning optimization framework in the synthesis of an API or a building block of an API, from screening to practical synthesis?

(Remarks on code availability)

Reviewer #3

(Remarks to the Author)

General Comments

The work by Sin, Schwaller, and co-workers presents an innovative approach to the optimization of nickel-catalyzed Suzuki reactions, combining high-throughput experimentation (HTE) with machine learning optimization frameworks. This study significantly advances the field by introducing a novel method for plate design for HTE, a critical topic with broad implications for in both academic and industry for synthetic chemistry and catalysis.

I strongly recommend publication in Nature Communications. Below are some specific comments which, I would like to see addressed before publication, however.

Specific Comments

- The authors should really be commended on this fantastic work, everything I would have liked to see explored in such a study (and more) has been investigated to a high degree. Clearly the authors have thought deeply about the problem, and come up with fantastic benchmarking to prove their work. The Supporting Information, contains everything I would expect and is well formatted. The GitHub repository is clean, and navigable, I was delighted to find all the raw data in it in a format conducive to further investigation.
- The only section I'm a little bit wary of is the comparison of the experimentalist designed HTE plates. The authors do a fantastic job in showcasing how the nickel precatalyst used is one of the most important factors in the reaction. However both experimentally designed plates hold the nickel precatalyst as a fixed variable. While, as the authors correctly note 'This underscores the limitations of traditional grid-like HTE plate designs, which are restricted in the diversity of reaction parameters explored in each plate and risk overlooking promising regions.'. I'm just a bit wary that if a chemist had just happened to have picked a better nickel source, the plate comparison in Fig 1. (a) might look less 'impressive' going from experimentalist designed to initial plate of ML optimization. This is however always an issue with making these comparisons and I don't necessarily disagree with the authors for including them. It might be interesting to see, if the experimentalists were given a second or third plate how their design evolves. My chemical intuition given the choice for a next plate would have been to investigate the nickel sources.

(Remarks on code availability)

The GitHub repository is clean, and navigable, all the raw data in it in a format conducive to further investigation is available.

Reviewer #4

(Remarks to the Author)

Although Schwaller and coworkers clearly demonstrate the scalability of Bayesian optimization campaigns by expanding the number of batched experimental suggestions from 16 to 96, the real question lies in the impact of this approach on the optimization efficiency.

The virtual benchmarks that evaluate the optimization efficiency of the q-NEHVI, qNParEgo, and TS-HVI acquisition functions (provided in the Supplementary Information Figures 3-5) do not demonstrate the advantage of increasing the batch size from 24 to 48 and 96 experiments. In fact, it appears that the larger batch sizes simply increase the number of experiments required to converge to the maximum % hypervolume in their most challenging optimization problem, the C-H arylation reaction (120 experiments with a batch size of 24, 240 experiments with a batch size of 48, and 384 experiments with a batch size of 96).

In my view, a higher batch size approach conflicts with the advantage of iterative Bayesian optimization, which is to converge on the global optimum and explore the search space with the highest level of efficiency. Instead the higher batch size approach appears to lead to an unnecessarily high number of experiments.

The impact conversation aside, developing a scalable Bayesian optimization strategy with a higher batch size per iteration is a technical improvement on current methods, however, the level of improvement does not warrant a publication in Nature Communications as currently framed.

I have two suggestions for reframing the impact of their optimization strategy:

The authors could carry out a virtual benchmarking study based on the data acquired from their nickel-catalyzed Suzuki system to virtually demonstrate the advantage of a higher batch size on optimization performance.

The authors could reframe their focus to the incorporation of PC descriptors from DFT-computed chemical features. The incorporation of chemical features in Bayesian optimizations can be challenging due to the significant expansion of the multidimensional search space. Perhaps their strategy is able to facilitate chemical feature incorporation more effectively than current methods.

(Remarks on code availability)

Version 1:

Reviewer comments:

Reviewer #1

(Remarks to the Author)

After reading the author's letter, the manuscript and the SI, I am satisfied with the authors responses to my criticisms on the experimental work (setup and analysis) and the storyline (additional use cases). While not all the experimental responses match my experience, it is now better documented in the SI, which will enable future practitioners to explore the methods to determine the best approaches.

As maintained from the start, I believe that the publishability of this work is mostly related to the quality of the ML application. If the ML expert reviewers believe that the work is novel, beneficial and correct, then I am happy to support publication from the experimental side.

(Remarks on code availability)

I did not review the code, as I have no experience therein.

Reviewer #2

(Remarks to the Author)

After reviewing the revised manuscript submitted by Sin et. al. titled "Highly Parallel Optimisation of Nickel-Catalysed Suzuki Reactions through Automation and Machine Intelligence" I am pleased to report that the author has successfully addressed all the comments and suggestions from my side.

The author's response was thorough and thoughtful, making the manuscript stronger and more impactful in the field of Machine Intelligence aided HTE. Given these improvements and the manuscript's overall high quality, I am confident that it meets the high standards of Nature Communications. Therefore, I strongly recommend that this manuscript be accepted for publication.

Thank you for the opportunity to review this submission. I believe it will be a valuable addition to our journal and will greatly benefit the HTE community.

(Remarks on code availability)

Reviewer #3

(Remarks to the Author)

I am happy with the changes, and I approve for publication in Nature Communications

(Remarks on code availability)

Reviewer #4

(Remarks to the Author)

The authors argue effectively in response my comment about increased batch size simply leading to a higher number of experiments by reframing the advantage of increased batch size around the reduction of temporal cost (calendar time) and iterative cost (number of iterations). Reframing the impact of their work in this way, in combination with the inclusion of additional benchmarking studies that highlight the advantage of increased batch size, alleviates my initial concern. The authors also do the manuscript more justice by highlighting the successful incorporation of reduced dimensionality DFT features in the optimization, which is a noteworthy aspect of their work. I recommend the manuscript to be published in its current form.

(Remarks on code availability)

Reviewer #1 (Remarks to the Author):

“The manuscript “Highly Parallel Optimisation of Nickel-Catalysed Suzuki Reactions through Automation and Machine Intelligence” by Sin et. al., details using automated HTE experimentation guided by autonomous ML protocols to optimise a Ni-catalysed Suzuki reaction. As expressed to the journal editors, as an expert in HTE chemistry, with no practical training in ML techniques, I can only weigh in on the experimental aspects of the work.”

Response 1a.

We thank the reviewer for providing their expertise on the experimental aspects of our work.

We would like to highlight an important aspect of our approach. In this work, our ML protocols were not operated autonomously or independently of human input. Rather, our work demonstrates the development and application of our ML-driven optimisation framework where we utilise ML to work in concert with experimentalist insights.

In our manuscript, experimentalist and user insights are explicitly integrated into several areas:

1. Defining meaningful chemical space boundaries based on practical process requirements and domain knowledge
2. Evaluating optimisation results within the broader context of practical process development goals to align and adjust optimisation strategies based on emerging results

We have demonstrated the success of the overall ML-driven optimisation framework through multiple experimental case studies, including additional revisions with **two experimental case studies of successful application to practical API process development at Roche**.

We have enhanced multiple sections of our manuscript to better highlight this synergistic aspect of our work, particularly in Section 2.1 (Optimisation pipeline), and Section 3 (Discussion).

Highlighted manuscript changes:

“We represented the reaction condition space as a discrete combinatorial set of potential conditions. These conditions comprised ~~composed of~~ reaction parameters such as reagents, solvents, and temperatures deemed plausible by a chemist for a given reaction, guided by practical process requirements and domain knowledge .

“After obtaining new experimental data, the chemist can choose to repeat this process for as many iterations as desired, usually terminating upon convergence, stagnation in improvement, or exhaustion of experimental budget. Throughout the campaign, the chemist can integrate evolving insights from each iteration with domain expertise and fine-tune the exploration-exploitation balance, or adapt the strategy to meet process specific development timelines.”

“By providing chemists with a clearer understanding of the ML-guided optimisation process and the rationale behind its suggestions, we can foster greater trust and adoption of ML optimisation approaches within the chemical community. We envision that such collaborative approaches between ML algorithms and human experts could substantially enhance the efficiency and effectiveness of reaction optimisation in academia and in pharmaceutical development.”

“I have thoughts on several aspects of the experimental work:

1.) Reaction choice. The choice of a Ni-catalysed Suzuki reaction for optimisation has positive and negative aspects. On the positive side, it is a very important transformation, especially in Process Chemistry, where the use of well-established Pd-catalysis is problematic for reasons described in the text. While it is a high-value goal to pursue this reaction with autonomous optimisation, I feel that it is a relatively niche reaction where the “expert-user” comparison is practically guaranteed to work, since most chemists have little practical experience to draw from. The rules of ligand/ base/ solvent choice for Ni reactions are far less generally understood than for the Pd-variant. Perhaps a description of the expert-users experience would be helpful here (i.e., they have significant experience optimising Ni-catalysed reactions). Even the literature does not have a strong representation of this type of reaction to draw from.”

Response 1b.

We thank the reviewer for their comments regarding our reaction choice and experimental design. We address both the context of our Ni-catalysed examples and our framework's applicability across more well-established reactions.

Choice of Ni-catalysed Suzuki reaction

We agree with the reviewer that a description of the expert-users' experience would be useful. Our chemists have extensive experience in metal-catalysed coupling reactions, particularly with traditional Pd-catalysed variants. This background reflects a common scenario in pharmaceutical process development, where teams navigate the transition from well-established Pd chemistry to emerging, sustainable, and high-value Ni-catalysed alternatives.

As the reviewer notes, Ni-catalysed Suzuki reactions are high-value transformations in pharmaceutical synthesis, despite being less well understood than their more problematic Pd-variants. This characteristic makes them compelling test cases for our ML-driven optimisation approach. The limited literature precedent and domain knowledge create precisely the type of challenging optimisation scenario where traditional approaches often struggle, yet such scenarios are increasingly common in modern process development as sustainable methods gain priority.

To demonstrate the broader applicability of our approach towards these novel transformations, we included a second Ni-catalysed Suzuki reaction example in active pharmaceutical ingredient (API) synthesis at Roche in our lab. We refer the reviewer to Section 2.7 (Application to active pharmaceutical ingredient (API) synthesis) and Section 2.7.1 (API Case study 1: Nickel-catalysed Suzuki reaction). In this example, our ML-driven optimisation workflow identified multiple reaction conditions with >99 AP yield and selectivity in just four iterations, contributing another successful experimental Ni-catalysed Suzuki case study. Importantly, chemical insights gained from both Ni-catalysed Suzuki examples in our manuscript allow chemists to actively develop expertise and build an understanding of this promising but less explored transformation.

By systematically exploring and mapping productive regions of the chemical space for these novel transformations, our ML-driven optimisation framework aids experimentalists in transforming these less well-understood reaction spaces (Ni-catalysis) into more well-established domains. This approach facilitates the development and uptake of these emerging, high-value transformations where precedent is limited, benefitting the broader chemistry community.

Additional of more reaction types

To demonstrate our ML protocol's broad utility, we have expanded our study beyond novel Ni-catalysed reactions to include established transformations. In Section 2.7.2 (API Case study 2: Pd-catalysed Buchwald-Hartwig reaction), we present a case study of a well-established Pd-catalysed Buchwald-Hartwig coupling in Roche API synthesis. In this example, our ML optimisation framework rapidly identified multiple reaction conditions achieving >95 AP yield and selectivity, leading to improvements upon the existing process conditions at scale. Notably, these optimised conditions were discovered in just 4 weeks, compared to the 6 months required for the original process development. This dramatic acceleration in optimisation timeline demonstrates our framework's ability to expedite pharmaceutical process development, even for well-understood transformations. These results underscore that even for established reaction classes with considerable precedent, our ML-guided workflow can reveal valuable opportunities for process improvement, complimenting existing chemical expertise.

“2.) HTE experimentation. I have several concerns with the experimental set-up as described in the supplementary information.

a. First, the experimentation is only very basically described. A more full description of the experimental set-up is required to understand the quality of the experiments. It appears that the experiments were set-up on an Unchained Labs robotics platform (from the text, not described in the SI). All solids were added to vials first, followed by liquid components. It would appear from the description that exceedingly small amounts of several components are added to the reactions (Ni pre-catalysts and ligands). In my experience, adding 0.2 mg of Ni pre-catalyst (3% loading) will have a very high error bar with a solid handling robot. As well, with each reaction having a unique solvent, there is time associated with dosing solvent at the end, which may lead to solvent evaporation and variable concentration. While these “errors-bars” might be acceptable in a “discovery phase” of optimisation- i.e., finding a workable starting point, a high error bar in subsequent continuous optimisation would lead to a high degree of variability, minimizing the ability to fine-tune conditions. Also, I feel that experimental sections of synthetic organic papers should be reproducible, so it is important to document how much of different reagents are added in reactions. Most robotics systems will provide a report on dosing, this should be included.”

Response 1c.

We appreciate the reviewer's detailed comments and constructive suggestions regarding experimental methodology. We have substantially expanded the SI to include extensive details on the experimental platform and all HTE (and verification experiments) to ensure reproducibility. We would be happy to add additional information if required.

Comments on solid dosing

Regarding solid dosing precision, our UnchainedLabs platform and other state-of-the-art-systems (e.g., Multidose from Labman, Quantos from Mettler-Toledo, and Swile from Chemspeed) can achieve accuracies ≤ 0.1 mg depending on the specific solid and dosing system.

We maintain target dispense values > 0.4 mg where possible. If smaller amounts are required, we utilise ChemBeads technology, enabling us to dispense small amounts of solids easily. In our Ni-catalysed Suzuki coupling experiments, we maintained dispense targets > 0.4 mg. The dispense targets for Ni-precursors ranged from 0.43 - 1.27 mg and ligands from 0.83 - 2.85 mg. The initialisation plate demonstrated high average accuracy and dispensing precision:

- ligands (± 0.11 mg, $\pm 9\%$)
- precursors (± 0.06 mg, $\pm 10\%$)
- substrates ($\pm 1.5\%$ and $\pm 0.5\%$)
- solid bases ($\pm 2.0\%$ and $\pm 0.2\%$)

These accuracies validate our microscale solid dispensing strategy, particularly given that our algorithm demonstrates robustness to experimental noise (Section 2.4) and can identify globally optimal solutions independent of some experimental variation, e.g. through dispensing.

Comments on solvent dosing

Regarding solvent dispensing, we agree that automated liquid handlers for chemistry are generally slow, especially with unique solvents in reaction wells. Thus, we specifically chose manual dispense strategies, using a Multipette E3 Single Channel Pipet, which allowed us to complete liquid addition within 5-10 minutes, minimising evaporation risks.

Overall, our high-throughput experimental approach achieves an optimal balance between throughput, experimental precision, material consumption, and cost. This approach is further enabled by our ML algorithm's demonstrated robustness to experimental variation.

“b (1). Second, as an experienced HTE experimentalist, I am concerned about the manner in which the experiments were conducted. In order to make the reaction components weighable (large enough quantities), they were conducted at 60 μmol scale, in 120 μL of a wide variety of solvents (non-polar to highly polar). Running at high concentration and with a variety of solvents will invariably lead to a high degree of reaction heterogeneity. The mixing approach is poorly described in the experimental section (“stirring disks”). Some validation (or a reference) where this method is effective at agitation at this scale and concentration would be important.”

Response 1d.

We value the reviewer's attention to experimental methodology. Our screening conditions are deliberately designed to align closely with those that are most similar to and are relevant for scale-up, using industrially relevant solvent volumes (5-20 vol) and heterogeneous bases. Our approach reflects practical considerations: higher concentrations are essential for viable reaction

rates and efficient scale-up, while heterogeneous bases typically offer superior activity, sustainability, and cost advantages in manufacturing settings over soluble bases. Screening under process-relevant conditions ensures alignment between optimisation and conditions that are practical at scale, streamlining overall process development.

As requested by the reviewer, we have added information regarding our mixing approach, covering detailed equipment specifications to Supplementary Information Section 5.1 (HTE platform), including information on screening plates and stirring disks. Our stirring disk setup (“Super Tumble Stir Disk”, from V&P Scientific) is widely adopted in HTE laboratories and has been extensively validated in our laboratory through:

1. Position independent results regardless of HTE plate locations (A1-H12)
2. Successful scale-up of identified conditions with this set-up from milligram to multi(hundred)-kilogram scale
3. Consistent performance across numerous optimisation campaigns in our laboratory

“b (2). In addition, the “mix it all together” approach is often a very poor way to form metal-ligand complexes. Catalyst pre-formation, and ideally, activation are tools that process chemists will invariably rely on to improve reaction performance. The issue with inadequate mixing and catalyst formation experiments is that the results will be enriched in the most robust experiments, rather than the reactions that have the best yield potential. Robustness is a critical aspect of process chemistry but can typically be built into process development once suitable conditions are identified. In this HTE optimisation experimentation, I believe the conditions should not have this robustness bias built in.”

Response 1e.

We appreciate the reviewer’s insights about catalyst formation strategies. We agree that “mix it all together” approaches have their drawbacks, and that efficient catalyst formation is important for the development of robust processes.

Our approach reflects several key considerations. Optimal catalyst formation and activation strategies are frequently precursor and ligand dependent. While general guidelines exist, it is not generally known a priori what approach will be most suitable for a specific precursor-ligand combination, particularly for less-explored reactions such as the Ni-Suzuki coupling. Different pre-formation strategies could introduce their own biases for different combinations.

Our experimental evidence also suggests that the “mix it all together” approaches we employed can be effective in identifying productive conditions. In our Ni-catalysed Suzuki-Miyaura reaction, premixing ligand and precursor ex-situ led to comparable results as the “mix it all”

approach in the initialisation plate. Similarly, while pre-mixing $[\text{Pd}(\text{allyl})\text{Cl}]_2$ with bulky biaryl phosphines ex-situ does not generally form $[\text{Pd}(\text{allyl})\text{Cl}(\text{PR}_3)]$ [1], these systems were among the best performers when “mixed all together” in our Pd-catalysed Buchwald-Hartwig example. Notably, productive reactions identified under this approach also led to performant reaction conditions at scale.

To further elaborate on our standard workflows, this initial screening phase as described in our manuscript, aims to identify promising reaction conditions (e.g. ligands, solvents, additives). These conditions then later progress into further optimisation, where various strategies such as catalyst formation/activation are explored to refine the most effective candidates for practical pharmaceutical processes. We have amended the manuscript to better articulate this staged approach, particularly in section 2.1 (Optimisation pipeline):

“Algorithmic exploration of these categorical variables enables identification of promising reaction conditions **combinations**, which **can guide** leads to further refinement **fine-tuning** of **continuous** parameters such as catalyst loading and **activation reaction-time** in later stages of the optimisation process.”

Our ML-driven optimisation framework can additionally be adapted to identify the most performant catalyst activation strategies:

1. Different catalyst activation strategies can be explicitly encoded as optimisation parameters, such that different activation strategies can be explored across different precatalyst/ligand combinations
 2. Our approach is not restricted to in-situ catalyst formation and can readily incorporate precatalysts, which are widely adopted in catalyst screenings
-

“c. Third, the analytical approach is odd to me. If the reaction are all giving the same product, then standard dilution of reactions should result in a product area count assessment, where the peak area is directly proportional to product formation. Given that then actually made and purified the product independently, it is possible to calculate the LC assay yield. Use of LC area percent is going to lead to lower quality data, as other UV-absorbing compounds (ligands for example) will lead to greater variability in the data. It should be possible to fix this retro-actively and report the data as actual yields, not LCAP.”

Response 1f.

We appreciate the reviewer's suggestion regarding the use of LC assay yield. While LC assay yield offers a valid approach for reaction comparison, we tend not to employ this strategy for two practical reasons:

1. It requires having a product reference standard before beginning optimisation - which is typically not the case in our process chemistry laboratory workflow. The product, though, is identified by MS.
2. It requires all reactions to have exactly the same volume and same amount of limiting substrate to be comparable. In practice, solvent evaporation and dispensing variance make this challenging to achieve.

“Use of LC area percent is going to lead to lower quality data, as other UV-absorbing compounds (ligands for example) will lead to greater variability in the data.”

Response 1g.

Regarding LC area percent measurements, we incorporate pre-processing steps to ensure accurate quantification. We utilise HTE OS [2] in our laboratory. HTE OS is integrated with a Spotfire application that enables tagging of all LCMS signals to species (e.g. “limiting SM”, “other SM”, “solvent”, “ignore peak”, etc.). All reactions are tagged accordingly. Importantly, UV-absorbing ligands and precatalyst peaks are tagged as “ignore peak”. All signals tagged as either “other SM”, “solvent” or “ignore peak” are then excluded and not used in the calculation of LCAP to ensure accurate calculations.

Our laboratory routinely employs either LCAP or calibration with an internal standard, added at the beginning of the reaction. For this study, we selected LCAP to minimise the impact of variations in limiting substrate and internal standard quantities. We have added detailed information about his data processing approach to the Supplementary Information Section 5.1 (HTE platform).

“d. Finally, I would like to see a scale-up of the best results identified.”

Response 1h.

As requested by reviewer #1 and #2, we have added further scaled-up experiments in the Supplementary Information for all of our ML-driven optimisation campaigns.

“3.) Comparison to expert guided work. One of the key experiments in this work is the comparison of autonomous optimisation to expert guided experiments. As mentioned above, limited expertise with Ni chemistry is likely to reduce the “expert’s” ability to come up with a good approach to solving this problem, relative to more well-used experiments. As well, the key finding, that Ni-pre-catalyst is the main determining factor, and that ligands are not important in the optimisation, is the key insight that enabled reaction optimisation. I believe that if the expert user was given this information, they could have readily optimised the reaction, especially with 4 additional rounds of HTE optimisation. What this suggests is that the expert user comparison should really be accomplished after each step to do an adequate comparison, I actually think that the next 3 rounds of autonomous optimisation, in which the algorithm chose to mostly run more exploratory experiments, rather than exploitative experiments (and the authors needed to change the protocol on the last step), suggests that the autonomous HTE experiments were not particularly well executed. In the end 79% yield after an initial ~50% yield in round 1, suggests that the additional 4 rounds of HTE were not particularly impactful. In the end, I don’t find the comparison particularly compelling, but this system did come up with a reasonable solution, albeit with some human intervention, which is commendable.

4.) Punchline. I feel that papers that describe new tools, particularly in high-end journals, should have a strong punchline that clearly demonstrates the value of the work. In my assessment, and again, without being able to weigh in on the ML aspects of the work, the punchline in this story is fairly weak. The most impactful aspects of the work appear to me to be the broad coverage of conditions using HTE in the first step (Sobol sampling), which enabled to identification of Ni pre-catalyst as the main driver in reaction performance. The additional rounds of autonomous, ML-driven experimentation did not appear to add a large amount of value. With a workable starting point, a typical chemist should be able to improve a reaction in similar or better efficiency. In the end, the more important point may be that the autonomous system was able to solve a difficult chemistry problem, albeit with some human intervention. This is the likely outcome of “autonomous approaches”, wherein people and computers will work together to solve

problems. In the end, if their new autonomous optimisation approach is truly enabling, they should be able to generate a more compelling story, in an efficient manner.

My opinion is that this work may have innovative aspects in terms of the ML optimisation techniques used, but comes up quite short of the mark in terms of the chemistry documentation, execution and the punch-line. In the end, if their new autonomous optimisation approach is truly enabling, they should be able to generate a more compelling story, in an efficient manner and with better documentation.”

Response 1i.

We appreciate the reviewer’s comments on our ML-guided optimisation framework and comparisons to traditional purely experimentalist-guided strategies. We would like to further elaborate on aspects of our work regarding the expert chemist comparison and our work’s impact, addressing reviewer #1 comments 3) and 4) together due to their overlapping themes.

“One of the key experiments in this work is the comparison of autonomous optimisation to expert guided experiments.

In the end, I don’t find the comparison particularly compelling, but this system did come up with a reasonable solution, albeit with some human intervention, which is commendable. ...

In the end, the more important point may be that the autonomous system was able to solve a difficult chemistry problem, albeit with some human intervention. This is the likely outcome of “autonomous approaches”, wherein people and computers will work together to solve problems. In the end, if their new autonomous optimisation approach is truly enabling, they should be able to generate a more compelling story, in an efficient manner.”

Response 1j.

As noted in our previous response (**response 1a.**), our ML-driven optimisation framework integrates ML capabilities with human expertise rather than operating autonomously. This synergistic approach enables comprehensive and accelerated exploration of reaction spaces compared to traditional optimisation approaches.

Our ML optimisation strategy employs Bayesian optimisation as a mathematical foundation for sequential decision making. While this protocol can operate autonomously, its parameters and optimisation behaviour can be actively steered according to the specific optimisation problem at hand. For example, our transition to exploitation-focused search in later optimisation stages reflects active decision-making by process chemists based on emerging results and specific practical timescale requirements. Such strategic decisions reflect our framework's flexibility in accommodating specific process development needs while maintaining systematic and efficient exploration of the chemical space.

“As mentioned above, limited expertise with Ni chemistry is likely to reduce the “expert’s” ability to come up with a good approach to solving this problem, relative to more well-used experiments.”

Response 1k.

As addressed in our prior response (**response 1b.**), while Ni chemistry is comparatively novel, such scenarios represent increasingly common practical scenarios where experimentalists transition to more sustainable and cost-effective approaches. Our ML workflow is effective across both emerging cases like Ni and widely-employed Pd-catalysed transformations, demonstrated by our experimental results.

“...As well, the key finding, that Ni-pre-catalyst is the main determining factor, and that ligands are not important in the optimisation, is the key insight that enabled reaction optimisation. I believe that if the expert user was given this information, they could have readily optimised the reaction, especially with 4 additional rounds of HTE optimisation.”

Response 1l.

We would like to provide several important clarifications to address the reviewers comments.

Parameter dependencies of optimisation outcome

While the optimal Nickel precatalyst $[\text{Ni}(o\text{Tol})\text{Cl}(\text{PPh}_3)_2]$ emerged as an important factor, our findings demonstrate that this knowledge alone does not enable reaction optimisation, highlighting the nuanced complexity of the reaction space.

1. Many reactions using the $[\text{Ni}(o\text{Tol})\text{Cl}(\text{PPh}_3)_2]$ precatalyst did not lead to any product formation, indicating that precatalyst choice alone does not determine reaction outcome.

Our SHAP analysis (Figure 5.) and reaction outcome distributions (SI Section 5.4) show that the choice of base, solvent, and co-solvent all substantially influence reaction success.

2. Our ML workflow successfully identified productive conditions (>40 AP yield) for the less performant precatalysts [Ni(COD)(DQ)] and [Ni(oTol)Cl(TMEDA)], whereas these catalysts showed no activity under the chemist-selected conditions. This highlights the importance of identifying optimal parameter combinations beyond precatalyst selection.

We have made multiple revisions to the manuscript in Section 2.6 (Application to Nickel-catalysed Suzuki reactions) to better articulate our optimisation process and post-campaign analysis.

Comparison to chemists

The reviewer suggests that “if the expert user was given this information (referring to mechanistic insights on the reaction), they could have readily optimised the reaction.” Regarding this assertion, suggesting optimisation would be straightforward with foreknowledge of chemical insights governing the reaction presents a retrospective bias.

The presence of an optimal Ni-precatalyst choice emerged as a key finding only after the completion of our ML-guided optimisation campaign, and was not known a priori. Our ML framework identified this chemical insight through systematic exploration of the chemical space, without being informed by the previous chemist-designed experiments. While these chemical insights may seem apparent in hindsight, the chemist's experimental design choices demonstrate that identifying this was non-trivial during the initial attempts with traditional optimisation approaches.

In our study, the chemist designed two sequential HTE batches, with knowledge of the first-batch results before designing the second. Despite iterative optimisation rounds and testing both [Ni(COD)(DQ)] and [Ni(oTol)Cl(TMEDA)] precatalysts, for which our ML workflow found productive conditions, traditional approaches identified no successful reactions. At this stage, given the exploration of various reaction conditions across two HTE batches, including different nickel sources, the chemist's conclusion that the reaction might be fundamentally challenging or unworkable would have been a reasonable assessment.

“... I actually think that the next 3 rounds of autonomous optimisation, in which the algorithm chose to mostly run more exploratory experiments, rather than exploitative experiments (and the authors needed to change the protocol on the last step), suggests that the autonomous HTE experiments were not particularly well executed. In the end 79% yield after an initial ~50% yield in round 1, suggests that the additional 4 rounds of HTE were not particularly impactful. In the end, I don't find the comparison particularly compelling, but this system did come up with a reasonable solution, albeit with some human intervention, which is commendable.

The most impactful aspects of the work appear to me to be the broad coverage of conditions using HTE in the first step (Sobol sampling), which enabled to identification of Ni pre-catalyst as the main driver in reaction performance. The additional rounds of autonomous, ML-driven experimentation did not appear to add a large amount of value.”

Response 1m.

We appreciate the reviewer's comments and would like to highlight several aspects and features of our ML-guided optimisation approach.

Clarification of ML-guided optimisation approach

As noted in previous responses (**response 1j.**), our ML framework employs Bayesian optimisation as its mathematical foundation, where the balance between exploration and exploitation is dynamically managed through ML-guided uncertainty quantification. While this can be operated autonomously, its parameters and optimisation behaviour can be actively steered by chemists, as was done in this work, to align with domain knowledge and process specific development requirements. The ability to shift between exploratory and exploitative strategies is a core feature of our framework that enables chemists to make active, informed judgements based on emerging results and practical timescale constraints.

The balance between exploration and exploitation naturally evolves during optimisation campaigns, guided by practical development context and requirements. While early exploration builds understanding of the reaction space, the decision to transition to focused exploitation depends on both emerging chemical insights and the specific reaction context. In our optimisation campaign, this approach led us to assess that only the fifth round warranted a more focused exploitation strategy, instead of throughout "the next 3 rounds".

We have made multiple revisions to the manuscript in Section 2.6 (Application to Nickel-catalysed Suzuki reactions), Supplementary Information Section 5.2.6 (Visualisation of experimental results) to better articulate our optimisation process and ML-driven optimisation framework.

Our framework's ability to transition to targeted exploitation phases demonstrated broad utility across two additional case studies in pharmaceutical API synthesis. Notably, in optimising a Pd-catalysed Buchwald-Hartwig reaction, a common and well-established transformation, our ML-guided optimisation framework identified multiple conditions with >95 AP yield and selectivity, leading to improved process conditions at scale in just 4 weeks compared to a previous 6-month campaign. Similarly, application to a second Ni-catalysed Suzuki reaction for API synthesis led to multiple conditions achieving >99 AP yield and selectivity within the same timeframe. These results highlight how our framework's flexibility - balancing systematic exploration with targeted exploitation - effectively addresses diverse reaction challenges while meeting pharmaceutical process development requirements.

Impact of optimisation

Regarding yield improvements in the Ni-catalysed Suzuki reaction, our ML-guided approach improved yields from 50% to 73% after a single ML guided iteration. However, the absolute magnitude of yield improvement should not be viewed in isolation and is best understood within its specific reaction context - different transformations present distinct chemical landscapes with varying optimisation potential. In this case, our first Ni-catalysed Suzuki reaction optimisation represents meaningful progress in a challenging reaction space where no productive conditions were identified through traditional approaches.

The broader value of our approach lies in its systematic identification of productive conditions across diverse reaction types. Our additional case studies demonstrate this capability, achieving highly optimised conditions (>95 AP yield and selectivity) while substantially accelerating optimisation timelines. Beyond yield improvements, this systematic exploration enables mapping and understanding less established reaction spaces, helping transform emerging transformations like Ni-catalysis into more well-understood domains through the identification of key chemical insights and productive regions of chemical space.

“What this suggests is that the expert user comparison should really be accomplished after each step to do an adequate comparison, ...

The most impactful aspects of the work appear to me to be the broad coverage of conditions using HTE in the first step (Sobol sampling), which enabled to identification of Ni pre-catalyst as the main driver in reaction performance. The additional rounds of autonomous, ML-driven experimentation did not appear to add a large amount of value. With a workable starting point, a typical chemist should be able to improve a reaction in similar or better efficiency.”

Response 1n.

We appreciate the reviewer's analysis which highlights several important aspects of our work, and would like to further elaborate on several important points:

As noted in previous responses (**response 1l.**), our analysis reveals the nuanced role of the Ni-precatalyst in optimisation success. While $[\text{Ni}(\text{oTol})\text{Cl}(\text{PPh}_3)_2]$ emerged as influential, reaction success depended on multiple parameters beyond precatalyst choice - many reactions using this precatalyst showed no activity. Additionally, our ML workflow identified productive conditions for the less performant precatalysts $[\text{Ni}(\text{COD})(\text{DQ})]$ and $[\text{Ni}(\text{oTol})\text{Cl}(\text{TMEDA})]$, whereas traditional approaches had failed to identify any, suggesting that unsuccessful experimental design did not stem from precatalyst choice alone.

The identification of the Ni-precatalyst as a key driver emerged from ML analysis of the complete optimisation campaign, and not from initial Sobol sampling alone. This insight was not apparent a priori, as demonstrated by the chemist's experimental choices across two iterations conducted with traditional approaches, including the deliberate testing of two different nickel sources without identifying productive conditions.

The reviewer suggests that "a typical chemist should be able to improve a reaction in similar or better efficiency" given a workable starting point. However, the identification of such starting points through traditional approaches often proves to be non-trivial. Our experimental evidence demonstrates this - the chemist-designed plates, despite incorporating iterative learning and systematic variation of nickel sources (for which our ML workflow identified productive conditions) across two rounds, did not identify any productive conditions. This outcome illustrates the non-trivial nature of the optimisation challenge.

We acknowledge that an alternative comparison could involve providing the Sobol sampling initialisation data to human experts as a starting point. However, our study specifically aimed to evaluate our ML-driven optimisation workflow, where Sobol sampling is an integral component, against current standard practices in process development. Our comparison to traditional approaches reflects the practical reality, where chemists traditionally rely only on their expertise and literature precedent to design HTE batches.

We have made multiple revisions to the manuscript in section 2.6 (Application to Nickel-catalysed Suzuki reactions) to better articulate our optimisation process and ML-driven optimisation framework.

Regarding the impact of ML-driven optimisation, we have addressed this in our previous responses (**response 1m.**).

“I feel that papers that describe new tools, particularly in high-end journals, should have a strong punchline that clearly demonstrates the value of the work. In my assessment, and again, without being able to weigh in on the ML aspects of the work, the punchline in this story is fairly weak.

...

My opinion is that this work may have innovative aspects in terms of the ML optimisation techniques used, but comes up quite short of the mark in terms of the chemistry documentation, execution and the punch-line. In the end, if their new autonomous optimisation approach is truly enabling, they should be able to generate a more compelling story, in an efficient manner and with better documentation.”

Response 1o.

We thank the reviewer for their constructive feedback on our chemistry documentation, which we have addressed through additions to the supplementary information and methods section (**response 1c-h**).

We would like to further elaborate on the impact and value of our work:

Our study presents a comprehensive demonstration of how our ML-driven HTE optimisation platform can accelerate pharmaceutical process development and address real-world challenges in chemical reaction optimisation, demonstrated through multiple experimental case studies.

First, we established its effectiveness in navigating challenging chemical space through the optimisation of a Ni-catalysed Suzuki reaction, where traditional approaches had failed to identify productive conditions. We further validated this capability through optimisation of a second Ni-catalysed Suzuki reaction in API synthesis, where our framework identified multiple conditions achieving >99 AP yield and selectivity. Importantly, our systematic exploration of these Ni-catalysed transformations revealed valuable chemical insights and mapped productive regions of the chemical space, facilitating the development and uptake of these emerging, high-value transformations in practical pharmaceutical synthesis where precedent is limited.

We also demonstrated our approach's broad utility through optimisation of a Pd-catalysed Buchwald-Hartwig coupling for API synthesis. Here, our framework identified multiple conditions achieving >95 AP yield and selectivity, which led to improved process conditions at gram scale which reduced a problematic impurity from 3.8% to <0.5% compared to the previous process while achieving >99% conversion. Most notably, our ML optimisation framework approach compressed a 6-month development timeline for the previous process to just 4 weeks

for the improved process, demonstrating dramatic acceleration in process optimisation even for well-established transformations, complimenting existing chemical expertise.

This practical impact is built on robust foundations. Our ML framework's development was guided by extensive computational benchmarks and statistical tests focusing on practical challenges in reaction optimisation, including scalability for integration with robotic equipment, experimental noise, batch constraints, and high-dimensional chemical spaces, facilitating robust performance in real-world applications.

Our ML protocol represents a practical, validated approach that combines systematic ML exploration with automation and expert knowledge to accelerate reaction optimisation. The consistent identification of superior conditions across diverse reaction types, coupled with acceleration of development timelines, validates its immediate practical impact in addressing real-world chemistry challenges.

References (Response to reviewer #1)

1. DeAngelis, A. J., Gildner, P. G., Chow, R., & Colacot, T. J. (2015). Generating active “L-PD(0)” via neutral or cationic Π -Allylpalladium complexes featuring Biaryl/Bipyrazolylphosphines: synthetic, mechanistic, and Structure–Activity studies in challenging Cross-Coupling reactions. *The Journal of Organic Chemistry*, 80(13), 6794–6813. <https://doi.org/10.1021/acs.joc.5b01005>
2. Wuitschik, G., Jost, V., Schindler, T., & Jakubik, M. (2024). HTE OS: A High-Throughput Experimentation Workflow Built from the Ground Up. *Organic Process Research & Development*, 28(7), 2875–2884. <https://doi.org/10.1021/acs.oprd.4c00160>

Reviewer #2 (Remarks to the Author):

“This article describes the development of a machine learning (ML) optimisation framework for batched multi-objective reaction optimisation, and overall strategy is well organized. The optimisation campaign identified reaction conditions with a 76% yield and 92% selectivity, which is a substantial improvement over the results obtained from traditional expert-designed high-throughput experimentation (HTE) plates that failed to yield successful reaction conditions. This showcased the practical value and effectiveness of the proposed approach in solving real-world reaction optimisation problems. The robustness of the framework to reaction noise and its ability to handle batch constraints encountered in real-world laboratories are also important contributions. This makes the approach more applicable in practical settings where such factors are inevitable.

In general, the optimisation framework developed here is innovative and practical, with good example described which further demonstrate some industrial applicability of this method.

However, here are some specific comments for this article, which are suggested to be taken care of by the authors before this article get published:

- The authors need to more clearly demonstrate the similarities and differences between the newly developed framework and existing approaches, i.e., q-NParEgo, q-NEHVI, TS-HVI. What’s the main improvements and advantages?”

Response 2a.

We appreciate the reviewer's request for clarification regarding our ML-driven optimisation framework. We elaborate on the key advantages provided by our framework and further mathematical and theoretical details of our methods Section 4.2 (Computational Implementation).

Enhanced computational stability

Multi-objective Bayesian optimisation in chemical reaction space is computationally challenging due to the presence of Pareto frontiers, where multiple solutions represent optimal trade-offs between competing objectives. Existing chemical reaction optimisation frameworks have used popular acquisition functions such as q-EHVI, facing significant computational limitations and leading to memory overflow due to computational complexity scaling exponentially with batch size. In our benchmarking, empirical studies show that these prior frameworks lead to memory bottlenecks on our GPU enabled workstation (see Supplementary Information section 2). These computational bottlenecks prevent effective integration with automated HTE workflows, particularly when scaling to the larger batch sizes common in modern robotic platforms. Thus, in the development of our ML optimisation framework, we sought to utilise more computationally

robust methods that overcome these computational bottlenecks, enabling efficient optimisation with larger experimental batches. Within these methods, scalability is achieved through cached box decompositions, single-sample approximations, and scalarisations, reducing computational load. As requested by the reviewer, we have amended the methods section (4.2) to provide detailed mathematical rationale for computational scalability of the methods evaluated in our study q-NParEgo, q-NEHVI, TS-HVI.

Noise robustness

As pointed out by reviewer #1, real-world experiments involve variability in experimental set-up and execution, thus leading to noise in observed reaction yields and selectivities. Our framework incorporates noise-robust acquisition functions that maintain optimisation performance even under significant experimental noise (section 2.4), validated through our computational experiments and statistical tests, making our approach more reliable in practical laboratory applications. Within the acquisition functions computation, noise robustness is achieved by integrating under possible realisations of true function given noisy observations. As requested by the reviewer, we have amended the methods section (4.2) to provide detailed mathematical rationale for noise handling of the methods evaluated in our study.

Integration with HTE workflows

To accommodate practical laboratory constraints, we developed constrained acquisition function optimisation and evaluation strategies, using computational benchmarks and statistical tests to identify the most effective approaches, providing robust foundations for practical, real-world application.

Exploitative framework

Our framework also incorporates a flexible exploitation strategy that can be activated to focus search efforts on promising regions once sufficient exploration has been achieved, allowing chemists to efficiently optimise identified leads.

“Although significant progress has been made in optimising this challenging nickel-catalysed Suzuki reaction through Automation and Machine Intelligence, it is notable that the optimal results obtained from the machine learning approach have not been transferred to benchtop for gram-scale validations and further scale-up, leading to some uncertainties in the practical value of the optimisation results generated from the framework.”

Response 2b.

As requested by reviewer #1 and #2, we have added further scaled-up experiments in the Supplementary Information for all of our ML-driven optimisation campaigns.

“The ML process started with Sobol sampling, aiming to maximize the coverage of the reaction space. Subsequent iterations involved model-guided Bayesian optimisation, and an exploitative optimisation strategy was implemented in the final iteration. Would 5 rounds of iterations sufficient for a standardized coupling reactions? If more rounds of iteration optimisations performed for the Suzuki case, would you expect higher yield of the desired product?”

Response 2c.

While the optimal number of rounds required for each different reaction (which may have different yield landscapes) cannot be determined precisely theoretically, we address the reviewers' comments with strong empirical evidence from multiple case studies.

We have supplemented our work with two additional examples of the application of our ML optimisation protocol applied to two Active Pharmaceutical Ingredients (API) synthesis. A second Ni-catalysed Suzuki reaction and a Pd-catalysed Buchwald-Hartwig reaction.

In both cases, multiple optimal reaction conditions (achieving >95 AP yield and selectivity) were identified within just 4 rounds of experiments. For our initial Ni-catalysed Suzuki reaction, AP yields of 70+ were already reached in the second round, and yields plateaued in later rounds, suggesting additional iterations would be unlikely to improve results within the given reaction space. Our computational studies on published benchmark datasets also support this observation, showing that convergence on optimal reaction conditions typically occurs within 5 rounds of experiments. Based on consistent evidence across experimental and computational studies, we conclude that 4-5 rounds are typically sufficient for optimising these coupling reactions.

“Although SHAP and other methods are used for feature importance analysis, the decision-making process of the machine learning model may still lack sufficient intuitiveness for process chemists in industry. How can the predictions of the machine learning model be further integrated with chemical intuition and domain knowledge? For example, could more explicit chemical explanations be provided to clarify why certain features (such as the nickel pre-catalyst) have a significant impact in the model and how these features are related to known chemical principles?”

Response 2d.

In our work, we demonstrated how our ML protocols can effectively identify important reaction parameters for complex and less well-understood reactions. We used SHAP feature importance analysis with experimental data collected from the optimisation campaign to identify the most impactful feature as the presence/absence of the optimal nickel precatalyst.

We chose one-hot encoding for precatalyst representation for two key reasons:

- The current lack of comprehensive, validated descriptor libraries for nickel precatalysts
- Literature evidence showing comparable performance between one-hot encoding and other more chemically rich representations [3, 4]

While one-hot encoding effectively guided optimisation, its binary nature (presence/absence of each precatalyst) limits mechanistic interpretation. Hence, our SHAP analysis can identify which precatalyst is important, but does not reveal which chemical properties (e.g. electronic effects, steric factors) drive performance. A more chemically intuitive representation using physical descriptors could enable SHAP to connect the ML insights to fundamental mechanistic understanding, assigning impact to specific descriptors.

The development of comprehensive and predictively validated descriptor representations for nickel precatalysts, informed by expert chemical knowledge, is an important direction for future research.

We appreciate the constructive comments provided by the reviewer, and we have expanded Section 3. (Discussion) to elaborate on integrating chemical insights with machine learning.

“The suggested improvements could be particularly effective when combined with explainable AI and visualisation methods, enhancing interpretability and decision-making. Furthermore, connecting explainable SHAP analysis with reaction-specific chemical descriptors could bridge ML predictions and fundamental chemical principles, enabling deeper mechanistic understanding. By providing chemists with a clearer understanding of the ML-guided optimisation process and the rationale behind its suggestions, we can foster greater trust and adoption of ML optimisation approaches within the chemical community. We envision that such collaborative approaches between ML algorithms and human experts could substantially enhance the efficiency and effectiveness of reaction optimisation in academia and in pharmaceutical development.”

“The article demonstrates the successful application in the nickel-catalysed Suzuki reaction. However, what is the generality of this optimisation framework in other types of chemical reactions? Have more extensive tests been conducted or is there a theoretical analysis to support its applicability in different reaction systems? Different reaction types may have diverse reaction kinetics, substrate characteristics, and optimisation requirements. The flexibility and effectiveness of the framework in the face of these variations need to be further explored.”

Response 2e.

We appreciate the reviewer's question about the generality of our optimisation framework. We would like to address this both practically and theoretically:

Practical Demonstration of Generality:

We have expanded our study to include two additional Active Pharmaceutical Ingredient (API) synthesis examples, where our ML optimisation protocol successfully identified optimal reaction conditions, achieving >95 AP yield and selectivity in both cases.

These additional examples span different reaction classes:

- A second Ni-catalysed Suzuki reaction, demonstrating reproducibility within the same reaction class
- A Pd-catalysed Buchwald-Hartwig reaction, showing applicability across more common metal catalysis systems

Our computational benchmarks further validate our approach, with our benchmark datasets using published reaction datasets from additional reaction types including Pd-catalysed Suzuki reactions and C-H arylation reactions, supporting our ML protocol's practical capability across a diverse range of reaction types. In internal projects at Roche, we also routinely apply our ML optimisation workflow successfully with transformations additionally including Pd-catalysed Suzuki-Miyaura Couplings, enolate arylations, Negishi couplings, and regioselective alkylations.

Theoretical analysis

From a theoretical perspective, different reaction types exhibit varying yield landscapes and kinetics thus leading to different relationships between reaction conditions and yield. Mathematically, these different reaction classes correspond to different underlying black-box functions between input reaction conditions and yield/selectivity.

Our ML protocol is built on Bayesian optimisation, a machine learning framework for optimising black-box functions that makes minimal assumptions about the underlying system, and does not

assume any functional form. Thus, our ML protocol does not rely on reaction-specific mechanistic assumptions, and is inherently adaptable to different reaction types.

Further, literature evidence of the successful application of other black box Bayesian optimisation methods to a wide range of reaction types [5, 6] suggest that many chemistry problems possess underlying functional relationships between reaction conditions and yield that satisfy the assumptions required for Bayesian optimisation to operate. [7]

“Is it possible for the authors to demonstrate a real industrial case of applying such learning optimisation framework in the synthesis of an API or a building block of an API, from screening to practical synthesis?”

Response 2f.

We have directly addressed the reviewer’s comments about industrial applicability by expanding our study to include real-world pharmaceutical applications.

We have supplemented our work with two additional case studies of our ML protocol applied to the synthesis of Active Pharmaceutical Ingredients (APIs) at Roche, demonstrating the practical utility of our ML optimisation protocol in an industrial context. We refer the reviewer to Section 2.7 (Application to active pharmaceutical ingredient (API) synthesis).

Our ML protocols applied to these industrial examples delivered significant practical value:

- Identification of multiple viable reaction conditions achieving >95 AP yield and selectivity for both APIs
- Discovery of improved process conditions at gram scale compared to previous process conditions for one API synthesis
- The examples span different reaction classes (Ni-catalysed Suzuki reaction and Pd-catalysed Buchwald-Hartwig reaction), demonstrating broad applicability in industrial settings

Due to commercial/IP considerations, the API structures are partially obscured, but represent real-world pharmaceutical drug synthesis applications rather than model systems.

References (Response to reviewer #2)

3. Pomberger, A., McCarthy, A. a. P., Khan, A., Sung, S., Taylor, C. J., Gaunt, M. J., Colwell, L., Walz, D., & Lapkin, A. A. (2022). The effect of chemical representation on active machine learning towards closed-loop optimization. *Reaction Chemistry & Engineering*, 7(6), 1368–1379. <https://doi.org/10.1039/d2re00008c>
4. Hickman, R. J., Ruža, J., Tribukait, H., Roch, L. M., & García-Durán, A. (2023). Equipping data-driven experiment planning for Self-driving Laboratories with semantic memory: case studies of transfer learning in chemical reaction optimization. *Reaction Chemistry & Engineering*, 8(9), 2284–2296. <https://doi.org/10.1039/d3re00008g>
5. Braconi, E., & Godineau, E. (2023b). Bayesian Optimization as a sustainable strategy for Early-Stage process Development? A case study of CU-Catalyzed C–N coupling of sterically hindered pyrazines. *ACS Sustainable Chemistry & Engineering*, 11(28), 10545–10554. <https://doi.org/10.1021/acssuschemeng.3c02455>
6. Dalton, D. M., Walroth, R. C., Rouget-Virbel, C., Mack, K. A., & Toste, F. D. (2024b). Utopia Point Bayesian Optimization finds Condition-Dependent selectivity for N-Methyl pyrazole condensation. *Journal of the American Chemical Society*, 146(23), 15779–15786. <https://doi.org/10.1021/jacs.4c01616>
7. Srinivas, N., Krause, A., Kakade, S.M., Seeger, M. (2009). Gaussian Process Optimization in the Bandit Setting: No Regret and Experimental Design. arXiv preprint arXiv:0912.3995.

Reviewer #3 (Remarks to the Author):

“General Comments

The work by Sin, Schwaller, and co-workers presents an innovative approach to the optimisation of nickel-catalysed Suzuki reactions, combining high-throughput experimentation (HTE) with machine learning optimisation frameworks. This study significantly advances the field by introducing a novel method for plate design for HTE, a critical topic with broad implications for in both academic and industry for synthetic chemistry and catalysis.

I strongly recommend publication in Nature Communications. Below are some specific comments which, I would like to see addressed before publication, however.

Specific Comments

- The authors should really be commended on this fantastic work, everything I would have liked to see explored in such a study (and more) has been investigated to a high degree. Clearly the authors have thought deeply about the problem, and come up with fantastic benchmarking to prove their work. The Supporting Information, contains everything I would expect and is well formatted. The GitHub repository is clean, and navigable, I was delighted to find all the raw data in it in a format conducive to further investigation.”

Response 3a.

We sincerely thank the reviewer for their positive feedback. We agree that releasing experimental data in a user-friendly format is crucial for advancing the field, and we hope our approach will facilitate further research in this area.

“The only section I’m a little bit wary of is the comparison of the experimentalist designed HTE plates. The authors do a fantastic job in showcasing how the nickel precatalyst used is one of the most important factors in the reaction. However both experimentally designed plates hold the nickel precatalyst as a fixed variable. While, as the authors correctly note ‘This underscores the limitations of traditional grid-like HTE plate designs, which are restricted in the diversity of reaction parameters explored in each plate and risk overlooking promising regions.’. I’m just a bit wary that if a chemist had just happened to have picked a better nickel source, the plate comparison in Fig 1. (a) might look less 'impressive' going from experimentalist designed to initial plate of ML optimisation. This is however always an issue with making these comparisons and I don’t

necessarily disagree with the authors for including them. It might be interesting to see, if the experimentalists were given a second or third plate how their design evolves. My chemical intuition given the choice for a next plate would have been to investigate the nickel sources.”

Response 3b.

We appreciate the reviewer's comments regarding the experimentalist-designed HTE plates comparison and welcome the opportunity for discussion on our experimental design:

As commented in previous responses (**response 11**), while the Ni-precatalyst emerged as an influential factor, our analysis revealed that successful optimisation required more than just precatalyst selection. Many reactions with the optimal $[\text{Ni}(\text{oTol})(\text{Cl})(\text{PPh}_3)_2]$ precatalyst showed no activity. Our ML workflow also identified productive conditions for the less performant precatalysts $[\text{Ni}(\text{COD})(\text{DQ})]$ and $[\text{Ni}(\text{oTol})(\text{Cl})(\text{TMEDA})]$, whereas traditional designs did not, suggesting the importance of identifying optimal parameter combinations beyond precatalyst selection.

In our study, the chemist designed two sequential HTE batches, with knowledge of the first-batch results before designing the second. Despite this iterative approach and testing the $[\text{Ni}(\text{COD})(\text{DQ})]$ and $[\text{Ni}(\text{oTol})(\text{Cl})(\text{TMEDA})]$ precatalysts for which our ML workflow found productive conditions, traditional approaches identified no successful reactions. At this stage, given the unproductive exploration of various reaction conditions with different nickel sources across two plates, concluding that the reaction was fundamentally challenging would have been reasonable.

As the reviewer notes, we agree that incidentally different initial precatalyst selections may have led to different comparative results, though our analysis shows that successful optimisation extends beyond identifying the optimal precatalyst. Indeed, this underscores non-trivial challenges in reaction optimisation - determining which parameters to prioritise among many for screening, and then identifying optimal combinations within a complex parameter space.

Our ML-guided approach addresses these challenges through systematic multi-parameter exploration of chemical space, efficiently revealing the key insights while simultaneously optimising other reaction parameters. This demonstrates its value in both accelerating reaction development and identifying opportunities that might otherwise have been overlooked by traditional approaches.

We have made multiple revisions to the manuscript in section 2.6 (Application to Nickel-catalysed Suzuki reactions) to better articulate the role of each reaction parameter in our experimental optimisation campaign.

Reviewer #4 (Remarks to the Author):

“Although Schwaller and coworkers clearly demonstrate the scalability of Bayesian optimisation campaigns by expanding the number of batched experimental suggestions from 16 to 96, the real question lies in the impact of this approach on the optimisation efficiency.

The virtual benchmarks that evaluate the optimisation efficiency of the q-NEHVI, qNParEgo, and TS-HVI acquisition functions (provided in the Supplementary Information Figures 3-5) do not demonstrate the advantage of increasing the batch size from 24 to 48 and 96 experiments. In fact, it appears that the larger batch sizes simply increase the number of experiments required to converge to the maximum % hypervolume in their most challenging optimisation problem, the C-H arylation reaction (120 experiments with a batch size of 24, 240 experiments with a batch size of 48, and 384 experiments with a batch size of 96).

In my view, a higher batch size approach conflicts with the advantage of iterative Bayesian optimisation, which is to converge on the global optimum and explore the search space with the highest level of efficiency. Instead the higher batch size approach appears to lead to an unnecessarily high number of experiments.

...

The authors could carry out a virtual benchmarking study based on the data acquired from their nickel-catalysed Suzuki system to virtually demonstrate the advantage of a higher batch size on optimisation performance.”

Response 4a.

We appreciate the reviewer's comments on optimisation efficiency and suggestions for highlighting the impact of our work. Following the reviewer's suggestion, we have conducted additional virtual benchmarking studies using our Ni-catalysed Suzuki reaction data and the additional API Pd-catalysed Buchwald Hartwig optimisation case study, which further demonstrate the benefits of larger batch sizes.

Temporal costs in pharmaceutical development

Our choice of batch size reflects the practical realities of pharmaceutical process development, where time-to-results (temporal cost) is often the critical factor. In our HTE laboratory, the cycle time from set-up to obtaining experimental results for a given HTE batch is approximately one week regardless of whether 24 or 96 experiments are run in the batch. Further, experimental effort remains comparable between 24-well and 96-well HTE plates. From this practical

perspective, where each iteration represents a week of development time, larger batch sizes enable four times more experimental throughput within the same timeline, crucial for meeting development timelines.

Advantages of larger batch size from benchmarking studies

We highlight that our original benchmarks demonstrated advantages of larger batch sizes, even in simpler search spaces (<8,000 reaction conditions). We have included additional figures in the Supplementary Information (Supplementary Figure 7) to better illustrate these results.

- As the reviewer notes, reflecting the reduced exploration needed in less complex parameter spaces, both large (96) and small (24) batch sizes achieved convergence within 5 iterations.
- However, larger batches showed superior performance (hypervolume %) at each iteration with faster convergence. Large batch sizes (96) consistently achieved convergence or near convergence within 2 iterations, while small batch sizes (24) required 4-5 iterations with variation between random seeds.
- Smaller batch sizes failed to achieve maximum hypervolume (%) (100 %) within 5 iterations in three out of four benchmarks.

These results translate directly to practical outcomes: larger batch sizes accelerate development timelines by reducing the number of experimental cycles required for optimisation.

Following the reviewers suggestion, we conducted additional benchmarking studies using optimisation data from our larger reaction condition search spaces - the Ni-catalysed Suzuki coupling (90,000 possible conditions) and Pd-catalysed Buchwald-Hartwig reaction (40,000 possible conditions) (Supplementary Information Figure 8). In these larger parameter spaces, the benefits of larger batches are further emphasised. Larger batch sizes (96 experiments) achieve comparable or better performance in half the iterations (practical cycle time) compared to smaller batches (24 experiments). Notably, smaller batch sizes also failed to achieve convergence on the API Pd-Buchwald benchmark. This marked difference in performance, particularly evident in large reaction search spaces, represents the complexity of real-world optimisation challenges, where broader parallel exploration becomes increasingly valuable as parameter space complexity grows.

“In my view, a higher batch size approach conflicts with the advantage of iterative Bayesian optimisation, which is to converge on the global optimum and explore the search space with the highest level of efficiency. Instead the higher batch size approach appears to lead to an unnecessarily high number of experiments.”

Response 4b.

We would like to elaborate on our approach by discussing the concept of efficiency and Bayesian optimisation in the context of our work.

Efficiency and cost in Bayesian optimisation

Bayesian optimisation (BO) is a probabilistic framework designed to efficiently navigate complex search spaces by reducing the prohibitive "cost" associated with exhaustive search methods. BO provides a principled method to finding high-quality locally optimal solutions while minimising cost.

Different dimensions of cost and efficiency

In optimisation problems, the "cost" that BO seeks to minimise can manifest in different dimensions. Which dimension of cost is most important depends on the specific application context.

- Sample cost: The total number of function evaluations (experiments) required
- Iterative & temporal cost: The number of sequential model-updates, experimental cycles and calendar time required to achieve optimisation

Optimisation efficiency in pharmaceutical process development contexts

As discussed in previous responses (**Response 4a.**), temporal cost (calendar time) and iterative cost (number of iterations) are the dominant factors in pharmaceutical process development. In our HTE platform, each experimental cycle requires one week regardless of batch size (24 or 96 experiments). This practical reality means larger batches enable four times more experimental throughput per unit time without proportionally increasing experimental effort.

Within this context, larger batch sizes enable the reduction of temporal cost by making each iteration more informationally efficient. As shown in our prior response (**Response 4a.**), our benchmarking studies consistently demonstrate this advantage. In simpler search spaces, larger batches achieved convergence in half the iterations (2 vs. 4-5), while in complex parameter spaces (Suzuki coupling and Buchwald-Hartwig reactions), this advantage became even more pronounced—with smaller batches failing to converge at all in some cases. This directly translates to weeks of saved development time in pharmaceutical process optimisation. The real-world benefit of our large batch Bayesian optimisation approach is further evidenced by our rapid optimisation of API synthesis (Section 2.7), where we compressed development timelines to just 4 weeks.

Similar batched approaches have been applied in other domains such as rover trajectory planning, vehicle structure optimisation, and optical display design [8, 9, 10], where large batched evaluations were necessary to address the high complexity of these design problems within a realistic timeframe. These fields, like chemical process development, involve navigating

large, high-dimensional solution spaces, making large batched evaluations crucial for achieving optimal outcomes in a reasonable time.

We have made revisions to the manuscript Section 1. (Introduction) to better articulate the importance of accelerated development timelines in pharmaceutical process development.

Highlighted manuscript changes:

“In the pharmaceutical industry, where rapid development is crucial, many reactions prove unsuccessful. There is a pressing need to expedite optimisation strategies in chemical synthesis beyond traditional approaches to meet increasingly demanding timelines in drug discovery and development. ~~where many reactions prove unsuccessful and timelines for drug discovery and development are continuously accelerating, there is a need to expedite optimisation strategies in chemical synthesis.~~ The natural synergy between ML optimisation and HTE platforms, leveraging efficient data-driven search strategies with highly parallel screening of numerous reactions, offers promising prospects for automated and accelerated chemical process optimisation in minimal experimental cycles.”

“The authors could reframe their focus to the incorporation of PC descriptors from DFT-computed chemical features. The incorporation of chemical features in Bayesian optimisations can be challenging due to the significant expansion of the multidimensional search space. Perhaps their strategy is able to facilitate chemical feature incorporation more effectively than current methods.”

Response 4c.

We agree with the reviewer regarding the technical challenges of handling high-dimensional search spaces in Bayesian optimisation of chemical reactions, which we view as one component of our study’s overall impact. As suggested by the reviewer, we have revised several sections (2.6 and 4.3) in our manuscript to emphasise the incorporation of dimensionally reduced chemical features, which is necessary to address the computational challenges associated with high-dimensional modeling using Gaussian Processes.

Highlighted manuscript changes:

“Given the large number of categorical variables, one-hot-binary encoding (OHE) featurisation was impractical. Instead, we employed DFT descriptors to represent ligands and solvents. To efficiently incorporate these high-dimensional chemical features, we leveraged dimensionality reduction techniques to enable effective exploration of the reaction space”

“We used 190 monophosphine ligand DFT descriptors from the Kraken featurisation workflow [51]. As high-dimensional representations require many data points for GP surrogate models to learn meaningful trends, we applied ~~applying~~ principal component analysis (PCA) to reduce the dimensionality of the ligand descriptor space ~~narrow down the 190 ligand descriptors from Kraken~~ to 37 principal components based on a 99% explained variance threshold, labeled ligand P C1 to ligand P C37. This allowed us to capture essential chemical information from ligands whilst maintaining a computationally tractable optimisation problem. Similarly, solvent descriptors were obtained from Moity et al. [52] with parameterisation using COSMOtherm and represented with 4 DFT-based descriptors labeled Solvent F 1 to Solvent F 4. The rest of the categorical variables (co-solvents, bases, and precatalysts) were featurised using OHE, with temperature remaining numerical.’

“The impact conversation aside, developing a scalable Bayesian optimisation strategy with a higher batch size per iteration is a technical improvement on current methods, however, the level of improvement does not warrant a publication in Nature Communications as currently framed.

I have two suggestions for reframing the impact of their optimisation strategy:

The authors could carry out a virtual benchmarking study based on the data acquired from their nickel-catalysed Suzuki system to virtually demonstrate the advantage of a higher batch size on optimisation performance.

The authors could reframe their focus to the incorporation of PC descriptors from DFT-computed chemical features. The incorporation of chemical features in Bayesian optimisations can be challenging due to the significant expansion of the multidimensional search space. Perhaps their strategy is able to facilitate chemical feature incorporation more effectively than current methods.”

Response 4d.

We thank the reviewer for their perspective on the framing of our work and their constructive suggestions for highlighting its impact. We have implemented both recommended suggestions. First, we have conducted additional virtual benchmarking studies using our Ni-catalysed Suzuki and Pd-catalysed Buchwald-Hartwig reaction data, which demonstrate clear advantages of larger batch sizes for accelerating process optimisation in addition to our existing benchmarks. Second, we have also revised sections emphasizing our handling of high-dimensional chemical feature spaces, highlighting the computational challenges of incorporating chemical descriptors in

Bayesian optimisation. While these technical aspects are important components of our work, we would additionally like to highlight how our study's impact extends beyond these considerations.

Our study presents a comprehensive demonstration of how our ML-driven HTE optimisation platform can accelerate pharmaceutical process development and address real-world challenges in chemical reaction optimisation, demonstrated through multiple experimental case studies.

First, we established its effectiveness in navigating challenging chemical space through the optimisation of a Ni-catalysed Suzuki reaction, where traditional approaches had failed to identify productive conditions. We further validated this capability through optimisation of a second Ni-catalysed Suzuki reaction in API synthesis, where our framework identified multiple conditions achieving >99 AP yield and selectivity. Importantly, our systematic exploration of these Ni-catalysed transformations revealed valuable mechanistic insights and mapped productive regions of the chemical space, facilitating the development and uptake of these emerging, high-value transformations in practical pharmaceutical synthesis where precedent is limited.

We also demonstrated our approach's broad utility through optimisation of a Pd-catalysed Buchwald-Hartwig coupling for API synthesis. Here, our framework identified multiple conditions achieving >95 AP yield and selectivity, which led to improved process conditions at gram scale which reduced a problematic impurity from 3.8% to <0.5% compared to the previous process while achieving >99% conversion. Most notably, our ML optimisation framework approach compressed a 6-month development timeline for the previous process to just 4 weeks for the improved process, demonstrating dramatic acceleration in process optimisation even for well-established transformations, complimenting existing chemical expertise.

This practical impact is built on robust foundations. Our ML framework's development was guided by extensive computational benchmarks focusing on practical challenges in reaction optimisation, including scalability for integration with robotic equipment, experimental noise, batch constraints, and high-dimensional chemical spaces, facilitating robust performance in real-world applications.

Our ML protocol represents a practical, validated approach that combines systematic ML exploration with automation and expert knowledge to accelerate reaction optimisation. The consistent identification of superior conditions across diverse reaction types, coupled with acceleration of development timelines, validates its immediate practical impact in addressing real-world chemistry challenges.

The impact of our work extends beyond optimisation to broader scientific contributions. Our systematic approach revealed valuable chemical insights into Ni-catalysed Suzuki reactions,

contributing to the understanding of these emerging sustainable transformations. Furthermore, we have released a comprehensive dataset of 1632 high-quality HTE reactions to benefit the broader research community.

References (Response to reviewer #4)

8. Daulton, S., Eriksson, D., Balandat, M., Bakshy, E. (2021). Multi-Objective Bayesian Optimization over High-Dimensional Search Spaces. arXiv preprint arXiv:2109.10964.
9. Wang, Z., Gehring, C., Kohli, P. & Jegelka, S.. (2018). Batched Large-scale Bayesian Optimization in High-dimensional Spaces. *Proceedings of the Twenty-First International Conference on Artificial Intelligence and Statistics, in Proceedings of Machine Learning Research* 84:745-754 Available from <https://proceedings.mlr.press/v84/wang18c.html>.
10. Kohira, T., Kemmotsu, H., Akira, O., & Tatsukawa, T. (2018). Proposal of benchmark problem based on real-world car structure design optimization. *Proceedings of the Genetic and Evolutionary Computation Conference Companion*. <https://doi.org/10.1145/3205651.3205702>